# Hierarchical Distribution Matching for Probabilistic Amplitude Shaping [note 1]

**DOI:** 10.3390/e22090958

**Published:** 2020-08-30

**Authors:** Stella Civelli, Marco Secondini

**Affiliations:** 1Tecip Institute, Scuola Superiore Sant’Anna, 56124 Pisa, Italy; 2Photonic Networks & Technologies National Laboratory, Consorzio nazionale interuniversitario per le telecomunicazioni (CNIT), 56124 Pisa, Italy; m.secondini@santannapisa.it

**Keywords:** probabilistic shaping, constellation shaping, distribution matcher, shaping gain, sphere shaping, constant composition, pulse position modulation, coded modulation

## Abstract

Probabilistic amplitude shaping—implemented through a distribution matcher (DM)—is an effective approach to enhance the performance and the flexibility of bandwidth-efficient coded modulations. Different DM structures have been proposed in the literature. Typically, both their performance and their complexity increase with the block length. In this work, we present a hierarchical DM (Hi-DM) approach based on the combination of several DMs of different possible types, which provides the good performance of long DMs with the low complexity of several short DMs. The DMs are organized in layers. Each upper-layer DM encodes information on a sequence of lower-layer DMs, which are used as “virtual symbols”. First, we describe the Hi-DM structure, its properties, and the encoding and decoding procedures. Then, we present three particular Hi-DM configurations, providing some practical design guidelines, and investigating their performance in terms of rate loss and energy loss. Finally, we compare the system performance obtained with the proposed Hi-DM structures and with their single-layer counterparts: a 0.19dB SNR gain is obtained by a two-layer Hi-DM based on constant composition DMs (CCDM) compared to a single-layer CCDM with same complexity; a 0.12dB gain and a significant complexity reduction are obtained by a Hi-DM based on minimum-energy lookup tables compared to a single-layer DM based on enumerative sphere shaping with same memory requirements.

## 1. Introduction

Over the last few years, probabilistic shaping techniques have been widely investigated to improve the performance and the flexibility of optical fiber networks. By assigning different probabilities to the constellation symbols, probabilistic shaping allows both to finely adapt the information rate to the channel signal to noise ratio (SNR) and to reduce the gap to the Shannon limit [1,2].

For a given symbol constellation, the probability distribution that maximizes the entropy for a constrained mean energy per symbol (or, conversely, that minimizes the mean energy per symbol for a given entropy) is the Maxwell–Boltzmann (MB) distribution [3], where each symbol is selected with a probability that depends on its energy.

An effective shaping approach, known as *probabilistic amplitude shaping* (PAS), employs a *distribution matcher* (DM) and a systematic forward error correction (FEC) encoder in the reverse concatenation configuration [4,5]. probabilistic amplitude shaping (PAS) allows to adjust the information rate and operate close to the Shannon limit over a wide range of channel SNRs, using  a fixed quadrature amplitude modulation (QAM) format, a fixed-rate FEC, and the pragmatic bit interleaved coded modulation scheme. Its seamless implementation in current coherent optical systems requires only the introduction of a distribution matcher (DM) at the transmitter and an inverse DM at the receiver, without any changes to the rest of the transceiver chain.

The key element of PAS is the DM, which maps *k* uniformly distributed bits (the information bits) to *N* probabilistically shaped amplitudes from the *M*-ary alphabet $A=\left\{1,3,\ldots ,2M-1\right\}$. At the receiver, the inverse DM (invDM) maps the shaped amplitudes back to the information bits. Basically, the DM establishes a map between the $2^{k}$ bit sequences of length *k* and $2^{k}$—among $M^{N}$—amplitude sequences of length *N*, which must be selected so that the amplitudes occur with the desired (target) probability distribution.

The possible solutions are not all equally good. In fact, inducing the target MB distribution on the transmitted symbols (amplitudes) is a necessary but not sufficient condition to achieve the ultimate shaping gain offered by PAS. In general, the DM introduces also a correlation between the *N* amplitudes of each block. This correlation reduces the actual entropy rate of the output process with respect to the case of i.i.d. amplitudes with the same probability distribution. In fact, while the generic output amplitude A∈A has a certain entropy H(A), which corresponds to the information rate that could be ideally encoded on i.i.d. amplitudes with that distribution, the generated sequence is correlated and carries a lower information rate R=k/N≤H(A). This rate reduction is known as the *rate loss*.

As an example, consider the alphabet A={1,3} and the target distribution p=(0.75,0.25), which gives an entropy H(A)=0.8113bits/amplitude. Now, consider the DM that associates k=2 bits to N=4 amplitudes as follows [5]
(1)$$\begin{array}{ccc} 00 & ⟷ & 1113\\ 01 & ⟷ & 1131\\ 10 & ⟷ & 1311\\ 11 & ⟷ & 3111 \end{array}.$$

The empirical distribution of the amplitudes in Equation (Equation 1) exactly matches the target distribution (one “3” and three “1” every four amplitudes). However, while the target distribution theoretically allows to encode up to 0.8113 bits/amplitude, the DM scheme in Equation (Equation 1) encodes only *R* = *k*/*N* = 0.5 bits/amplitude, with a relevant rate loss of 0.3113 bits/amplitude. This rate loss is caused by the correlation between the amplitudes generated by the DM. A short block length usually imposes strict constraints on the output sequence—for instance, the sequence generated by a repeated use of Equation (Equation 1) will never contain more than six “1” or two ”3” in a row—inducing a strong correlation between the amplitudes. On the other hand, a longer block length may relax these constraints and reduce the rate loss.

Ideally, a good DM should have the following characteristics: be invertible; be fixed-to-fixed-length to avoid the drawback of a variable transmission rate [3] and the risk of buffer overflow; generate symbols with the desired target distribution (e.g., the MB distribution); generate almost i.i.d. (with as few constraints as possible) symbols to minimize the rate loss; have low implementation complexity.

A possible DM type that fulfill all these requirements reasonably well is *constant composition distribution matching* (CCDM) [6], in which the composition (i.e., the number of occurrences of each symbol of the alphabet) of the output sequence is fixed to have the desired target distribution, whereas information is encoded on the order of the *N* symbols. For instance, the DM in Equation (Equation 1) is a (very short) constant composition distribution matching (CCDM). A quite simple implementation of CCDM based on arithmetic coding has been proposed in Reference [6]. The rate loss of CCDM and the divergence from the target distribution vanish when the block length *N* goes to infinity. A good performance is practically achieved with a block length that typically ranges from several hundreds to a few thousands of symbols. Unfortunately, also the complexity increases (linearly) with the block length, so that alternative approaches are being investigated.

CCDM is a good approach, but is not optimal. For a given block length *N*, the most efficient way to encode *k* information bits is to map their $2^{k}$ possible realizations to the $2^{k}$ minimum-energy sequences of *N* amplitudes. This approach is known as *sphere shaping*: representing the sequences of *N* amplitudes in an *N*-dimensional space, the minimum-energy sequences are all contained within a sphere of given radius. For large *N*, sphere shaping induces (as expected, given its optimality in terms of energy efficiency) an MB distribution on the symbols of the alphabet. Moreover, the rate loss decreases with *N* (and vanishes for N→∞) significantly faster than for CCDM. A straightforward implementation of this method can be obtained through a look up table (LUT) containing the $2^{k}$ minimum-energy sequences. The use of a LUT avoids the need of complex encoding (and decoding) operations. However, the memory required to store the $2^{k}$ sequences increases exponentially with the block length *N*, so that this approach is feasible only for short block lengths, with a large rate loss. A more efficient implementation of sphere shaping for long block lengths is *enumerative sphere shaping* (ESS) [7,8], which uses a bounded-energy trellis to map binary indexes to amplitude sequences (and back). The memory required to store the trellis is still significant (it grows cubically with *N*), but much smaller than that required by the LUT. Moreover, it requires performing some operations between large numbers, with a complexity that grows linearly with *N* (as in CCDM).

Alternative methods for realizing a fixed-to-fixed length DM have been proposed, trying to obtain a better trade-off between performance, complexity, and memory [8,9,10,11,12]. Some of these methods operate by combining several simple DMs into a more complex DM with a better performance. For instance, *multiset-partition DM* combines several CCDMs with different compositions, addressed by a prefix, to expand the number of addressable sequences, reducing the rate loss [10]. *hierarchical DM* (Hi-DM) [9,12] combines several small LUTs in a hierarchical structure, to obtain an efficient DM with a reasonable memory requirement.

Elaborating on the latter ideas, we have recently proposed a generalized hierarchical DM (Hi-DM) approach that combines several short DMs of any kind in a hierarchical structure, in which the DMs of each layer are used as “virtual symbols” by the DMs of the upper layer [13]. In addition to the information encoded by the DMs of the bottom layer, each upper layer encodes some information on the sequence of DMs selected in the lower layer. This additional information is encoded “for free”, that is, without transmitting additional amplitudes. With a proper selection of the hierarchical structure and of the component DMs, the proposed Hi-DM approach can provide the small rate loss of a long DM with the small computational cost and memory requirements of the short component DMs. In this work, we expand on Reference [13] by providing a detailed description of the Hi-DM structure and by studying some specific configurations, their design, their performance, and their complexity. In particular, we show how to combine different DM types to obtain hierarchical structures based on code position modulation, CCDM, or LUTs. For each structure, we provide some simple examples, discuss the computational complexity and required memory, and compute the reduction of rate loss and energy loss obtained with respect to the equivalent-complexity single-layer structures. Finally, we compare the overall system performance that can be achieved with the proposed Hi-DM structures and with previously proposed CCDM [6] and enumerative sphere shaping (ESS) [7]  structures.

The manuscript is organized as follows. Section 2 provides some preliminary concepts and some useful performance metrics. Section 3 introduces the Hi-DM approach, describing the main idea and the general structure, illustrating the encoding and decoding procedures, and providing some simple examples. Three specific types of Hi-DM—namely, hierarchical code position modulation (Hi-CPM), hierarchical CCDM (Hi-CCDM), and hierarchical LUT (Hi-LUT)—are respectively studied in Section 4, Section 5 and Section 6, investigating their performance (in terms of rate and energy loss) and hardware requirements, and providing some useful design guidelines. In Section 7, the proposed Hi-DM structures are compared to the equivalent-complexity single-layer CCDM [6] and ESS [7] in terms of achievable information rate with bitwise decoding over the AWGN channel. The conclusions are finally drawn in Section 8.

## 2. Preliminaries

In the following, vectors are described with bold letters, and their components with italic and subscript. For example, the *ℓ*th component of the vector b is $b_{\ell }$.

A DM maps a vector $b=(b_{1}\ldots ,b_{k})$ of *k* uniformly distributed bits to a vector $A=(A_{1},\ldots ,A_{N})$ of *N* probabilistically shaped amplitudes from the *M*-ary alphabet $A=\left\{1,3,\ldots ,2M-1\right\}$
(2)$$\begin{array}{cc} \mathrm{DM}:{\left\{0,1\right\}}^{k} & \to T\subseteq A^{N}\\ b=(b_{1}\ldots ,b_{k}) & \mapsto A=(A_{1},\ldots ,A_{N}), \end{array}$$
with T being the set of possible output vectors selected by the DM. The DM should be *invertible*: the invDM maps the shaped amplitudes back to the information bits. In the following, two DMs, ${\mathrm{DM}}_{1}$ and ${\mathrm{DM}}_{2}$, are said to be *disjoint* if their respective codomains, $T_{1}$ and $T_{2}$, are disjoint. In general, a DM can be designed to approximate any target distribution. However, when the goal is to maximize the information rate for a given SNR over the additive white Gaussian noise (AWGN) channel, the target becomes the MB distribution [3].

The DM induces a certain probability distribution on the output amplitudes. Let the random variable *A* represent the generic amplitude at the output of the DM, the probability that A=x∈A is  [14]
(3)$$p\left(x\right)=\frac{\#\;\mathrm{of}\;\mathrm{occurrences}\;\mathrm{of}\;x\;\mathrm{in}\;T}{2^{k}N}.$$

The entropy of *A* is [14]
(4)$$H\left(A\right)=-\sum_{x\in A}p\left(x\right)\log_{2}p\left(x\right).$$

By contrast, the actual rate of the DM (in bits per amplitude) is
(5)$$R=\frac{k}{N}.$$

Since the amplitudes at the output of the DM are not independent, the actual information rate *R* is always lower than the entropy H(A) of each amplitude. The difference is the rate loss
(6)$$R_{\mathrm{loss}}=H\left(A\right)-R\geq 0,$$
which is widely used as a performance metric to compare different DM approaches. Note that, for a given target rate $R_{0}$ and block length *N*, the actual rate is $R=\frac{k}{N}=\frac{1}{N}\left\lfloor R_{0}N\right\rceil$. In the following, the rate loss is the difference between the entropy and the actual rate, rather than the target rate.

The efficiency of the DM can be considered also from an energy perspective. For a given DM with rate *R*, the mean energy per amplitude is
(7)$$E_{\mathrm{DM}}=\sum_{x\in A}p\left(x\right)x^{2}.$$

On the other hand, the same rate could be ideally achieved with mean energy $E_{\mathrm{MB}}$, corresponding to the mean energy of the MB distribution with entropy *R*. The energy loss of the DM is defined as the ratio (usually expressed in dB)
(8)$$E_{\mathrm{loss}}=\frac{E_{\mathrm{DM}}}{E_{\mathrm{MB}}}.$$

For instance, the small DM in Equation (Equation 1) has rate R=0.5bits/amplitude and mean energy $E_{\mathrm{DM}}=\left(3\times 1^{2}+3^{2}\right)/4=3$. On the other hand, the mean energy of the MB distribution with entropy 0.5 bits/amplitude is $E_{\mathrm{MB}}=1.8802$. Therefore, the energy loss of the DM in Equation (Equation 1) is $E_{\mathrm{loss}}=2.0292\;dB$.

Often, the rate loss and the energy loss of a DM can be seen as the two sides of the same coin. This is true when the probability distribution of the output amplitudes is (with a good approximation) the MB distribution, with respect to which the energy loss is defined. In this case, minimizing the rate loss is tantamount to minimizing the energy loss, meaning that the the two quantities can be used interchangeably as a performance metric. On the other hand, they are not equivalent when the output amplitudes have not an MB distribution—this may happen as a deliberate choice or as a consequence of the design procedure. For instance, considering again the example in Equation (Equation 1), if we replace all the occurrences of the amplitude “1” with “3” and vice versa, we obtain a new DM with same rate *R* = 0.5 bits/amplitude, same entropy *H*(*A*) = 0.8113 bits/amplitude, but higher mean energy $E_{\mathrm{DM}}=7$. Thus, the new DM has the same rate loss $R_{\mathrm{loss}}$ = 0.3113 bits/amplitude as Equation (Equation 1), but a much higher energy loss $E_{\mathrm{loss}}$ = 5.7089 dB. In this case, the rate loss is clearly not a good metric to compare the energy efficiency of the two DMs.

The previous example shows that (i) the DM design is essential to achieve good results—very short or badly designed DMs may have a poor performance; (ii) the rate loss and the energy loss can be very high even if the DM exactly matches the target MB distribution; and (iii) the rate loss and energy loss are not equivalent when the output DM distribution is not the target MB distribution. In the following, we will consider both the rate loss and the energy loss as performance metrics, verifying if they provide consistent results in terms of optimization and performance comparison.

An alphabet of *M* amplitudes—or levels—yields a 2M pulse amplitude modulation (PAM) constellation when considering the sign of the symbols, and a $4M^{2}$-QAM constellation when considering both the sign and the in-phase and quadrature components. In the first case, the rate will be 1+R, while in the second case it will be 2(1+R). The rate loss Equation (Equation 6) is the rate loss of the DM, while the rate loss induced on the QAM constellation—considered in other works, for example, References [9,12]—is doubled.

## 3. Hierarchical Distribution Matcher

The efficiency and the complexity of a DM typically depend on its block length. A short block length requires only few computational resources but entails a significant rate and energy loss. On the contrary, a long block length allows to reduce the rate and energy loss, but requires more computational resources. The Hi-DM approach tries to attain a better trade-off between complexity and performance by combining several short DMs in a hierarchical structure with a longer overall block length. The Hi-DM structure should simultaneously provide a low rate and energy loss thanks to the long overall block length, and a low complexity thanks to the short block length of each component DM.

The main concept is explained here for a 2-layer structure. Given a set of short DMs, $\left\{D_{1}^{\left(1\right)},\ldots ,D_{M_{1}}^{\left(1\right)}\right\}$, the goal is to combine them to obtain a longer DM. A simple option is that of using the DMs sequentially, in a prescribed order, concatenating their output. In this case, though the resulting structure can be seen as a single longer DM, the overall rate and mean energy per symbol are just the (weighted) average of the rates and energies of the individual DMs, with no improvement on the overall performance. On the other hand, the set of DMs $\left\{D_{1}^{\left(1\right)},\ldots ,D_{M_{1}}^{\left(1\right)}\right\}$ can be collected in a first layer and used by a second-layer DM $D^{\left(2\right)}$ as a sort of “virtual alphabet”: $D^{\left(2\right)}$ encodes some information bits by selecting the specific sequence of DMs to be used in the first layer. Then, as in the previous option, each DM used in the first layer encodes information at its own rate, and the resulting mean energy per symbol is a (weighted) average of the mean energies required by the individual DMs. However, in this case, the overall rate is increased thanks to the additional bits that are encoded “for free” (i.e., without transmitting additional symbols) by the second layer on the virtual symbols (the DMs).

The Hi-DM structure can be generalized to multiple layers, and can employ different DM types (e.g., ESS, CCDM, …) in each layer, as shown in the following sections. In particular, the LUT-based Hi-DM proposed in References [9,12] can be seen as a special case of the general Hi-DM structure considered in this work. The strength of a particular Hi-DM—in terms of both rate loss and complexity—depends on the characteristics (type, block length, rate) and combination of the component DMs.

In the rest of this section, we provide two toy examples to illustrate the Hi-DM concept, we define the general structure and the notations, we describe the encoding and the decoding algorithms, and, finally, we give some basic rules to determine the main properties (i.e., rate, output probability, entropy, and energy) of the Hi-DM structure.

### 3.1. Toy Examples

#### 3.1.1. Toy Example 1

In the first toy example, we consider a simple 2-layer Hi-DM—in the first layer, we have two disjoint DMs, $D_{1}^{\left(1\right)}$ and $D_{2}^{\left(1\right)}$, respectively with mean energy $E_{1}^{\left(1\right)}$ and $E_{2}^{\left(1\right)}$, both with $k_{1}=3$ input bits, $N_{1}=2$ output shaped amplitudes, and rate $R_{1}=3/2$—their actual type is irrelevant in this example.

The two DMs are combined to obtain a longer DM with 4 output amplitudes. Figure 1 shows three possible configurations. In configuration (a), the two DMs are used sequentially and independently, in a deterministic order: $D_{1}^{\left(1\right)}$ always takes the first three bits and produces the first two amplitudes (respectively, 111 and 13 in this example), while $D_{2}^{\left(1\right)}$ takes the next three bits and produces the next two amplitudes (010 and 53, respectively). Overall, this non-layered approach encodes six bits over four amplitudes, with rate R=6/4=3/2 and mean energy $E=(E_{1}^{\left(1\right)}+E_{2}^{\left(1\right)})/2$—no improvement is observed with respect to the case of a single DM.

In configuration (b), the two DMs are collected in the first layer, and the order with which they are used is selected in the second layer, depending on the value of one additional information bit. If the first bit is 0, as in the example shown in Figure 1b, the first permutation $D_{1}^{\left(1\right)}D_{2}^{\left(1\right)}$ is selected; if the bit is 1, the other permutation $D_{2}^{\left(1\right)},D_{1}^{\left(1\right)}$ is selected. The next six bits are encoded as before, three by each DM (in the selected order), producing the four output amplitudes. In this manner one additional bit is encoded on the same four amplitudes. The rate is increased to R=7/4, while the mean energy is unchanged, $E=(E_{1}^{\left(1\right)}+E_{2}^{\left(1\right)})/2$. In this case, the invDM decodes the first bit by recognizing from which DM the two couples of output amplitudes have been generated—this is possible because the DMs are disjoint.

Eventually, configuration (c) improves on configuration (b) by allowing all the combinations with repetitions of the two DMs, as sketched in Figure 1c. In this case, the first two bits (01 in this example) are used to decide which DM sequence is used out of the four ones listed in the second layer. Also in this example, the sequence $D_{1}^{\left(1\right)}D_{2}^{\left(1\right)}$ is selected. The last six bits are then encoded as before by the two selected DMs, producing the four output amplitudes. Thus, the overall rate is further increased to R=8/4, whereas the mean energy remains unchanged, $E=(E_{1}^{\left(1\right)}+E_{2}^{\left(1\right)})/2$, as each DM is used with probability 1/2.

To better appreciate the potential gain offered by the hierarchical approach, we can make the simplifying (but non-necessary) assumption that the two component DMs have the same mean energy. In this case, all the three configurations in Figure 1 have the same mean energy per symbol as each component DM, $E=E_{1}^{\left(1\right)}=E_{2}^{\left(1\right)}$. However, while configuration (a) has also the same rate, the hierarchical configurations (b) and (c) achieve higher rates. Thus, according to Equation (Equation 8), they reduce the energy loss compared to the individual DMs.

#### 3.1.2. Toy Example 2

In the second toy example, we consider the particular case in which all the DMs are implemented by LUTs. Suppose that we want to encode k=5 bits on N=3 amplitudes from the alphabet $\left\{1,3,5,7\right\}$ of M=4 elements, with rate R=k/N=5/3≈1.6667. In this case, the optimum (minimum-energy) DM can be implemented by the small LUT in Figure 2a, whose entries contain the $2^{k}=32$ lowest-energy sequences of three amplitudes. The five-bit input sequence determines the entry of the LUT in which the corresponding three-amplitude output sequence is contained.

According to Equations (Equation 3)–(Equation 7), this DM has output probabilities p(1)=0.3854, p(3)=0.3125, p(5)=0.2292, p(7)=0.0729, which correspond to an entropy H=1.8171 and a mean energy per symbol E=12.5. By contrast, the mean energy per symbol of an MB distribution with entropy 5/3 is 9.6418, which corresponds to the minimum energy per symbol required by an ideal (infinitely long) DM with the prescribed rate R=5/3. Thus, the DM considered in Figure 2a has rate loss $R_{\mathrm{loss}}=1.8171-1.6667=0.1504$ and energy loss $E_{\mathrm{loss}}=10\log_{10}\left(12.5/9.6418\right)=1.1275\;dB$.

On the other hand, Figure 2b shows a two-layer Hi-DM structure with the same overall rate R=5/3, but made of several small LUTs with the same block length as the LUT in Figure 2a. The LUTs are designed to minimize the mean energy per symbol, according to the general procedure detailed in Section 6.1. In the long-block-length limit, this would correspond to an MB target output distribution. The first (or lower) layer comprises $M_{2}=3$ disjoint DMs, namely $D_{1}^{\left(1\right)}$, $D_{2}^{\left(1\right)}$, and $D_{3}^{\left(1\right)}$. Each DM encodes $k_{1}=4$ bits on $N_{1}=3$ output amplitudes. The amplitudes are drawn from the alphabet $\left\{1,3,5,7\right\}$ of $M_{1}=4$ elements. The three DMs are implemented by LUTs. The LUT entries are filled by ordering all the possible $M_{1}^{N_{1}}=64$ output sequences according to their mean energy; the first $2^{k_{1}}=16$ (lowest-energy) sequences are assigned to $D_{1}^{\left(1\right)}$, the second 16 sequences to $D_{2}^{\left(1\right)}$, the third 16 sequences to $D_{3}^{\left(1\right)}$, and the last (highest-energy) 16 sequences are discarded. The mean energies per symbol of the three DMs are computed by averaging the energies of their respective output sequences, obtaining $E_{1}^{\left(1\right)}=7.8333$, $E_{2}^{\left(1\right)}=17.1667$, and $E_{3}^{\left(1\right)}=24$. The second (or upper) layer is again a LUT, denoted as $D^{\left(2\right)}$, which encodes $k_{2}=3$ bits on $N_{2}=3$ output virtual symbols drawn from the alphabet $D^{\left(1\right)}=\left\{D_{1}^{\left(1\right)},D_{2}^{\left(1\right)},D_{3}^{\left(1\right)}\right\}$ of $M_{2}=3$ elements (the DMs of the first layer). Also this LUT is designed by ordering all the $M_{2}^{N_{2}}=27$ possible output sequences according to their mean energy—for example, the second sequence $D_{1}^{\left(1\right)}D_{1}^{\left(1\right)}D_{2}^{\left(1\right)}$ has mean energy $(E_{1}^{\left(1\right)}+E_{1}^{\left(1\right)}+E_{2}^{\left(1\right)})/3=10.9444$—and using the first $2^{k_{2}}=8$ sequences to fill the entries of the LUT. The mean energy per symbol of the overall structure is $E=E_{2}=11.7986$, which is obtained by averaging the mean energy per symbol of the output sequences of $D^{\left(2\right)}$. Simple calculations allow to compute also the output probabilities of the three first-layer DMs and, considering the occurrences of each DM in the second layer, also the overall output probabilities of the Hi-DM structure and the corresponding entropy H=1.7878.

An encoding example is shown in Figure 2c. The input bit sequence is 011000010100001. The first $k_{2}=3$ bits, 011, are sent to the upper DM $D_{2}$, causing the selection of the fourth sequence, $D_{2}^{\left(1\right)}$$D_{1}^{\left(1\right)}$$D_{1}^{\left(1\right)}$. Thus, the following $N_{2}=3$ blocks of $k_{1}=4$ bits, 0000, 1010, and 0001, are sent to $D_{2}^{\left(1\right)}$, $D_{1}^{\left(1\right)}$, and $D_{1}^{\left(1\right)}$, respectively, which generate the three amplitude sequences 531, 511, and 113. The corresponding decoding procedure is illustrated in Figure 2d. The amplitude sequence is divided into blocks of $N_{1}=3$ amplitudes; each block is used to determine which DM in $D^{\left(1\right)}$ generated it, obtaining the sequence $D_{2}^{\left(1\right)}$$D_{1}^{\left(1\right)}$$D_{1}^{\left(1\right)}$. Thus, the inverse of each DM from the latter sequence is used to decode each block of amplitudes and obtain the corresponding $k_{1}=4$ bits. Moreover, the sequence of DMs is decoded using the inverse of $D_{2}$, producing the bits 011. In this case, the sequence can simply be found inside its LUT. A more general discussion about finding the DM sequences is reported in Section 3.4.

Overall, the Hi-DM structure described above is characterized by the vectors k=(4,3), N=(3,3), and M=(4,3)—formally defined in Section 3.2 and denoting, respectively, the number of input bits, output symbols, and alphabet elements of the two layers—and encodes $k=k_{2}+N_{2}k_{1}=15$ bits on $N=N_{1}N_{2}=9$ amplitudes, with a rate R=15/9≈1.667, a mean energy per symbol E=11.7986, and an entropy H=1.7878. Thus, with respect to the single-layer structure considered in Figure 2a—the best possible with a block length N=3—the Hi-DM structure in Figure 2b achieves the same rate with lower energy and entropy. Indeed, the rate loss is reduced to $R_{\mathrm{loss}}=1.7878-1.6667=0.1211$, and the energy loss to $E_{\mathrm{loss}}=10\log_{10}\left(11.7986/9.6418\right)=0.8767\;dB$.

### 3.2. Structure and Notation

A generic Hi-DM structure with three layers is sketched in Figure 3, the extension to more layers is straightforward. The encoding and decoding procedures are reported in the next sections. The  notation is described below.

The layers are indicated with ℓ=1,…,L, where ℓ=1 denotes the lowest layer, which produces the final amplitudes, whereas ℓ=L denotes the uppermost layer. The DMs in the *ℓ*th layer are indicated as $D_{h}^{\left(\ell \right)}$, for $h=1,\ldots ,M_{\ell +1}$ (see below for $M_{\ell +1}$). If the subscript is not reported, $D^{\left(\ell \right)}$  indicates the union of all the DMs in the *ℓ*th layer.$N_{\ell }$ is the number of output symbols (amplitudes for ℓ=1, DMs for ℓ>1) from each DM in the *ℓ*th layer.$k_{\ell }$ is the number of uniformly distributed input bits to each DM in the *ℓ*th layer.$M_{\ell }$ is the size of the alphabet used by the DMs of the *ℓ*th layer, that is, the DM $D_{h}^{\left(\ell \right)}$ takes values in an alphabet of $M_{\ell }$ elements. Each element is represented by $m_{\ell }=\left\lceil \log_{2}M_{\ell }\right\rceil$ bits. The number of output bits from each $D_{h}^{\left(\ell \right)}$ is $N_{\ell }m_{\ell }$.The *ℓ*th layer has $M_{\ell +1}$ DMs, $D_{1}^{\left(\ell \right)},\ldots ,D_{M_{\ell +1}}^{\left(\ell \right)}$, which constitute the alphabet of the layer above ($M_{L+1}=1$, since there is always a single DM in the top layer). Each DM produces one out of $2^{k_{\ell }}$ possible sequences of $N_{\ell }$ elements from an alphabet of order $M_{\ell }$.The *ℓ*th layer is used $T_{\ell }=\prod_{h=\ell +1}^{L}N_{h}$ times, with $T_{L}=1$.The number of output amplitudes, that is, the overall block length, is $N=T_{0}=\prod_{\ell =1}^{L}N_{\ell }$.The overall number of input bits is $k=\sum_{\ell =1}^{L}k_{\ell }T_{\ell }=\sum_{\ell =1}^{L}k_{\ell }\prod_{h=\ell +1}^{L}N_{h}$.The overall rate is
(9)$$R=\frac{k}{N}=\sum_{\ell =1}^{L}\frac{k_{\ell }}{\prod_{h=1}^{\ell }N_{h}}=\sum_{\ell =1}^{L}\frac{R_{\ell }}{\prod_{h=1}^{\ell -1}N_{h}},$$
with $R_{\ell }$ being the rate of each DM in the *ℓ*th layer.

To ensure that the resulting Hi-DM structure is invertible and fixed-to-fixed-length, we will make the following assumptions:All the DMs of a given layer *ℓ* are *disjoint*, that is, they do not have any common output sequences.All the DMs of a given layer *ℓ* have the same block length $N_{\ell }$.All the DMs of a given layer *ℓ* take the same number of input bits $k_{\ell }$.

The combination of Conditions 1 and 2 ensures that the Hi-DM is (easily) invertible, whereas the combination of Conditions 2 and 3 ensures that it is fixed-to-fixed-length and implies that all the DMs of layer *ℓ* have the same rate $R_{\ell }=k_{\ell }/N_{\ell }$. Note that these conditions are sufficient but not necessary to guarantee the desired properties. In fact, for the special case of CCDM-like component DMs considered in Section 5, we will be able to remove condition 3 without loosing the fixed-to-fixed-length property.

A Hi-DM structure is uniquely defined by the vectors M, N, k, and by the nature of the component DMs (they can be LUT, CCDM, etc.). The parameters M, N, and k can be varied at will to adjust the rate, the complexity, and the performance, with the constraint
(10)$$M_{\ell +1}2^{k_{\ell }}\leq M_{\ell }^{N_{\ell }},\;\ell =1,\ldots ,L.$$

This constraint ensures that, at each layer, the number of possible output sequences $M_{\ell }^{N_{\ell }}$ is sufficient to fill the $2^{k_{\ell }}$ entries of each of the $M_{\ell +1}$ disjoint DMs. The same rate can be obtained with different choices of M, N, and k, possibly resulting in different performance and complexity.

**Remark** **1.**
*As required by Condition 3 above, we assumed that all the DMs of the ℓth layer have the same number of input bits $k_{\ell }$. In principle, a different number of input bits could be considered for each DM, without affecting the invertibility of the code. This, however, would not guarantee that the overall Hi-DM is fixed-to-fixed-length. For example, let us consider a two-layer structure with the same $D^{\left(2\right)}$ as in Figure 2b, but with a different number of input bits $k_{1,h}$ to each DM $D_{h}^{\left(1\right)}$ of the first layer. In this case, the total number k of input bits depends on the sequence selected in the upper layer: it will be $k=k_{2}+3k_{1,1}$ if the first row of $D^{\left(2\right)}$ is selected, $k=k_{2}+2k_{1,1}+k_{1,2}$ if the second row is selected, and so on. On the other hand, having the same $k_{1,h}=k_{1}$, for h=1,2,3, will ensure that the number of input bits is fixed, $k=k_{2}+3k_{1}$, regardless of the selected row. Note, however, that this is a sufficient but non-necessary condition. For instance, it is not required when the component DMs are CCDM, as it will be shown in Section 5. In this case, we will allow each DM $D_{h}^{\left(\ell \right)}$ to have its own number of input bits, which will be denoted as $k_{\ell ,h}$.*


### 3.3. Encoding

The Hi-DM structure consists of several component DMs. The way in which each DM maps its input bits to the corresponding output symbols depends on the specific DM implementation (e.g., CCDM, ESS, LUT) and is not discussed here [6,7]. On the other hand, the way in which the component DMs are combined to obtain the overall encoding procedure is illustrated by Algorithm 1 in pseudocode notation.
**Algorithm 1** Hi-DM encoding.1:**Data:** Input bits $b=(b_{1},\ldots ,b_{k})$2:**Result:** The output amplitude sequence $d^{\left(1\right)}$3:initialization $d^{\left(L+1\right)}=D^{\left(L\right)}$4:**for**ℓ=L,…,1**do**5: **for**
$h=1,\ldots ,T_{\ell }$
**do**
6:  Take $b_{t}$ the first $k_{\ell }$ bits of b;7:  Remove $b_{t}$ from b;8:  Let $D_{j}^{\left(\ell \right)}$ be the DM in the component $d_{h}^{\left(\ell +1\right)}$9:  Use $D_{j}^{\left(\ell \right)}$ to map $b_{t}$ to a train of $N_{\ell }$ symbols and obtain the train of symbols $d_{\left(h-1\right)N_{\ell }+1}^{\left(\ell \right)},\ldots ,d_{hN_{\ell }}^{\left(\ell \right)}$, where $d_{i}^{\left(\ell \right)}\in \left\{D_{1}^{\left(\ell -1\right)},\ldots ,D_{M_{\ell }}^{\left(\ell -1\right)}\right\}$ if ℓ>1 and $d_{i}^{\left(\ell \right)}\in \left\{1,3,\ldots ,2M_{1}-1\right\}$ if ℓ=1;10:  Append the train of symbol $d^{\left(\ell \right)}=\left(d^{\left(\ell \right)},d_{\left(h-1\right)N_{\ell }+1}^{\left(\ell \right)},\ldots ,d_{hN_{\ell }}^{\left(\ell \right)}\right)$;11: **end for**12:**end for**

The word *symbol* indicates a DM when ℓ>1 and an amplitude {1,3,…} when ℓ=1. The procedure starts from the top layer, which takes the first $k_{L}$ input bits and generates the sequence of $N_{L}$ symbols (DMs), $d_{1}^{\left(L\right)},\ldots ,d_{N_{L}}^{\left(L\right)}$, with $d_{h}^{\left(L\right)}\in \left\{D_{1}^{\left(L-1\right)},\ldots ,D_{M_{L}}^{\left(L-1\right)}\right\}$. This sequence of DMs is then used in the (L−1)th layer to map the next $N_{L}k_{L-1}$ bits to the $N_{L}N_{L-1}$ symbols (DMs) to be used in the next lower layer—each DM mapping $k_{L-1}$ bits to $N_{L-1}$ symbols. The method is then iterated for the remaining layers, ℓ=L−2,…,1, $T_{\ell }$ being the number of total symbols generated by the *ℓ*th layer. In particular, when the first (lowest) layer is reached, the sequence of $T_{1}=\prod_{\ell =2}^{L}N_{\ell }$ DMs selected by the second layer maps the last $T_{1}k_{1}$ input bits on the $N=T_{0}=T_{1}N_{1}$ output amplitudes from the alphabet $\left\{1,3,\ldots ,2M_{1}-1\right\}$—each DM mapping $k_{1}$ bits on $N_{1}$ amplitudes. An example is shown in Figure 2c.

### 3.4. Decoding

The decoding procedure reverses the encoding operations and is reported in Algorithm 2 in pseudocode notation. Also in this case, the inverse decoding for each component DM (e.g., CCDM, ESS, LUT) is given for granted, so that we focus only on the overall procedure that combines the component DMs. Moreover, a specific discussion on the “DM recognition” step needed for decoding is reported at the end of the section.

The procedure starts from the first layer. The received amplitude sequence, of length $T_{0}=N=\prod_{\ell =1}^{L}N_{\ell }$, is divided into $T_{1}$ subsequences of length $N_{1}$. For each subsequence, the corresponding DM from which it has been generated is found in the set $\left\{D_{1}^{\left(1\right)},\ldots ,D_{M_{2}}^{\left(1\right)}\right\}$, obtaining the sequence of DMs $d_{1}^{\left(1\right)},\ldots ,d_{T_{1}}^{\left(1\right)}$— details about this DM recognition step are provided below. The corresponding sequence of invDMs is then used to decode the $T_{1}$ subsequences and obtain the last $T_{1}k_{1}$ bits of the original input bit sequence. Moreover, the sequence $d_{1}^{\left(1\right)},\ldots ,d_{T_{1}}^{\left(1\right)}$ is passed to the second layer, where the same decoding procedure is repeated: the sequence is divided into $T_{2}$ subsequences of length $N_{2}$; each subsequence is associated to the corresponding DM and decoded by the corresponding invDM, obtaining the previous $T_{2}k_{2}$ bits of the original sequence and the new DM sequence $d_{1}^{\left(2\right)},\ldots ,d_{T_{2}}^{\left(2\right)}$ to be passed to the next layer. The decoding procedure is iterated until the top layer is reached and the first $k_{L}$ bits of the original sequence are decoded. An example is shown in Figure 2d.
**Algorithm 2** Hi-DM decoding.1:**Data:** Received amplitude sequence $d^{\left(1\right)}=\left(d_{1},\ldots ,d_{N}\right)$2:**Result:** The *k* decoded bits b3:**for**ℓ=1,…,L**do**4: Divide $d^{\left(\ell \right)}$ into $T_{\ell }$ sequences of length $N_{\ell }$$d^{\left(\ell \right)}=\left(a_{1},\ldots ,a_{T_{\ell }}\right)$5: **for**
$h=1,\ldots ,T_{\ell }$
**do**
6:  Find *j* such that $a_{h}$ is produced by $D_{j}^{\left(\ell \right)}$.7:  Decode $a_{h}$ using $D_{j}^{\left(\ell \right)}$ and obtain $k_{\ell }$ bits $b_{t}$.8:  Append the bits $b=(b_{t},b)$9:  If ℓ<L, append the sequence of DM $d^{\left(\ell +1\right)}=\left(d^{\left(\ell +1\right)},D_{j}^{\left(\ell \right)}\right)$10: **end for**
11:**end for**

At each layer, for each received subsequence, the decoding algorithm has to find the corresponding DM that generated it. This DM is unique, since all the DMs on the same layer are disjoint. This operation can be implemented in different ways, depending on the particular kind of DMs employed in the layer. Though finding the most efficient implementation is outside the scope of this paper, we provide here some useful indications for the cases that will be considered in the next sections.

When several CCDMs with different compositions are employed, the correct CCDM composition can be determined by counting the occurrences of each symbol (DM or amplitude) in the subsequence.

When several DMs with different energy levels are employed—for example, $D_{1}^{\left(\ell \right)}$ produces sequences with energy $E\leq E_{1}$, $D_{2}^{\left(\ell \right)}$ produces sequences with energy $E_{1}<E\leq E_{2}$, and so on—the correct DM can be determined by computing the energy of the subsequence. This can be done independently of the particular DM implementation, which, for instance, could be based on the ESS algorithm or use a LUT.

More in general, when arbitrary LUTs are employed for encoding, DM recognition and inversion can be jointly performed in each layer by a single LUT that inverts $D^{\left(\ell \right)}$, that is, the union of all the DMs of the layer. Given a sequence of $N_{\ell }$ symbols from an alphabet of order $M_{\ell }$, the inverse LUT returns the corresponding $k_{\ell }$ decoded bits, plus $m_{\ell +1}$ bits denoting which DM from the set $\left\{D_{1}^{\left(\ell \right)},\ldots ,D_{M+1}^{\left(\ell \right)}\right\}$ has generated the sequence. This approach is quite general, as it does not rely on any specific property of the component DMs (e.g., constant composition, energy, ...), and seems equivalent to the method proposed in References [9,12]. In terms of lookup time, the most efficient implementation is based on index mapping (also known as trivial hash function), in which the symbol sequence is directly used as the LUT address, so that no computations and a single LUT access are required. However, index mapping is not memory-efficient, as it requires storing a LUT with $M_{\ell }^{N_{\ell }}$ rows, one per each possible sequence in the output space. On the other hand, according to Equation (Equation 10) some memory can be saved by storing only $M_{\ell +1}2^{k_{\ell }}$ rows, one per each sequence effectively addressed by the $M_{\ell +1}$ DMs of the layer. In this case, a slightly more complex but still effective lookup strategy can be employed, for example, by storing the sequences in lexicographic order and using a binary search algorithm.

### 3.5. Probability Distribution

For a given Hi-DM structure with *L* layers, the output probability p(x) of $x\in \{1,3,\ldots ,2M_{1}-1\}$ is evaluated recursively—using the law of total probability [14]—as
(11)$$p_{\ell }\left(x\right)=\sum_{y\in \{1,2,\ldots ,M_{\ell +1}\}}p_{\ell |\ell +1}\left(x|y\right)p_{\ell +1}\left(y\right),$$
where $p_{\ell }\left(x\right)$ is the probability of the symbol *x* after layers L,L−1,…,ℓ, and
(12)$$p_{\ell |\ell +1}\left(x|y\right)=\frac{\#\;\mathrm{of}\;\mathrm{occurrences}\;\mathrm{of}\;x\;\mathrm{in}\;D_{y}^{\left(\ell \right)}}{2^{k_{\ell }}N_{\ell }},$$
is the probability that the symbol *x* is generated in the *ℓ*th layer, given that *y* was chosen in the (ℓ+1)th layer. The recursion starts from the top layer with
(13)$$p_{L}\left(x\right)=\frac{\#\;\mathrm{of}\;\mathrm{occurrences}\;\mathrm{of}\;x\;\mathrm{in}\;D^{\left(L\right)}}{2^{k_{L}}N_{L}},$$
and ends at the first layer with $p\left(x\right)=p_{1}\left(x\right)$.

The entropy and mean energy per symbol of the output amplitudes can be easily evaluated from the output probability using Equations (Equation 4) and (Equation 7), though a direct recursion for the mean energy will also be provided in Section 6.1.

## 4. The Hi-Dm with Code Position Modulation (CPM): Hierarchical CPM

In this Section, we describe a Hi-DM structure that, in analogy with the pulse position modulation concept [15], encodes information on the position of a given (higher-energy) DM in a sequence of many identical low-energy DMs. We will denote this structure as Hi-CPM. Let us assume to have a DM with good performance. In a single-layer implementation, this DM is used many times to encode blocks of bits on blocks of symbols. In the proposed Hi-CPM structure, the idea is to replace, from time to time, this DM with another one, disjoint from the first one. Even if the performance of the second DM might not be as good as that of the first one, the gain is provided by the fact that some additional information bits can be encoded on the position of the second DM, similarly to how information is encoded on the position of a pulse in pulse position modulation.

### 4.1. Principle of Operation

The first layer comprises *L* disjoint DMs $\left\{D_{1}^{\left(1\right)},\ldots ,D_{L}^{\left(1\right)}\right\}$ of any kind, with decreasing performance and the same rate $k_{1}/N_{1}$. Often, an MB target distribution is considered, with the goal of minimizing the energy for a given rate. In this case, the DMs are sorted in increasing order of energy. We start by describing a two-layer structure. In this case, only two DMs are required. The second layer generates a sequence of $N_{2}=2^{k_{2}}$ DMs, selecting $N_{2}-1$ times the first DM, and once the second DM, whose position is used to encode $k_{2}=\log_{2}\left(N_{2}\right)$ information bits. Therefore, each DM encodes $k_{1}$ bits on $N_{1}$ amplitudes, while $k_{2}$ additional bits are encoded every block of $N_{2}$ DMs. Overall, $k=k_{1}N_{2}+k_{2}$ bits are encoded on a block of $N=N_{2}N_{1}$ amplitudes.

To add a third layer, one considers $N_{3}=2^{k_{3}}$ repetitions of the two-layer structure, one of which is modified by replacing the second DM with the third one. The position of the modified structure is used to encode $k_{3}=\log_{2}\left(N_{3}\right)$ additional bits. With respect to the two-layer structure, $k_{3}$ additional bits are encoded every $N_{3}$ uses of the two-layer structure, with an overall rate of $k=k_{3}+N_{3}k_{2}+N_{2}N_{3}k_{1}$ bits on $N=N_{3}N_{2}N_{1}$ amplitudes. An example for L=3 is shown in Figure 4a. Here, for the sake of simplicity, the first layer considers the “trivial” DM—the DM $D_{\ell }^{\left(1\right)}$ encodes $k_{1}=0$ bits and produces the amplitude 2ℓ−1. The structure uses $D_{3}^{\left(1\right)}$ once, $D_{2}^{\left(1\right)}$$N_{3}-1$ times, and $D_{1}^{\left(1\right)}$ the remaining $N_{3}\left(N_{2}-1\right)$ times.

The concept can be easily extended to *L* layers, using *L* disjoint DMs. The DM $D_{\ell }^{\left(1\right)}$ is used $S_{\ell }$ times, with $S_{\ell }=T_{\ell }-T_{\ell +1}=\left(N_{\ell +1}-1\right)T_{\ell +1}$ for ℓ<L and $S_{L}=1$. A total of $k=\sum_{\ell =1}^{L}k_{\ell }T_{\ell }$ bits are encoded on $N=\prod_{\ell =1}^{L}N_{\ell }$ amplitudes.

The most interesting feature of Hi-CPM lays in the extreme simplicity of the encoding and decoding of the upper layers (ℓ=2,…,L), which respectively consist in using information bits as DM addresses and the other way round, with almost no additional complexity with respect to the single DMs used in the first layer.

The Hi-CPM concept can be further extended by selecting, at each layer, more than one low-energy DM to be replaced by higher-energy DMs, in analogy to the multi-pulse position modulation concept [16]. Unfortunately, this would, in general, increase the encoding and decoding complexity, to account for the relative positions of the modified DMs. In practice, each layer would work as a CCDM, encoding information on a particular permutation of the DMs. However, for the special case of two modified DMs—referred to as Hi-CPM2 in the following—a simple trick can be used to keep the same simple encoding and decoding procedures used in Hi-CPM. For each block of length $N_{\ell }$, two groups of $\log_{2}\left(N_{\ell }\right)$ bits are used to select the positions of two DMs. Then, to distinguish the order with which the two DMs have been selected (which is required to obtain a correct decoding at the receiver), one *check bit* is generated, with value 1 if the first pulse is located after the second, and 0 otherwise. This check bit is combined with other $2\log_{2}\left(N_{\ell }\right)-1$ information bits to obtain the next two groups of bits to be encoded.

If the two groups of bits are identical, they correspond to the same address. In this case, just one DM is modified and the check bit becomes irrelevant and can be arbitrarily set. At the *ℓ*th layer, for ℓ≥3, each selected address corresponds to a sequence of DMs, containing either one or two higher-level DMs $D_{\ell -1}^{\left(1\right)}$. In the first case, the DM is upgraded to $D_{\ell }^{\left(1\right)}$. In the second case, only the first DM is upgraded to $D_{\ell }^{\left(1\right)}$, as this is enough to ensure a correct decoding. An example of this procedure is shown in Figure 4b, where orange colored bits are the additional check bits that do not carry any information.

The advantage of Hi-CPM2 over Hi-CPM can be understood by considering one block of $2N_{\ell }$ DMs in the former case, and two blocks of $N_{\ell }$ DMs in the latter. For the same overall length, Hi-CPM encodes $2\log_{2}\left(N_{\ell }\right)$ bits ($\log_{2}\left(N_{\ell }\right)$ per block), whereas Hi-CPM2 encodes $2\log_{2}\left(2N_{\ell }\right)-1=2\log_{2}\left(N_{\ell }\right)+1$ bits ($\log_{2}\left(2N_{\ell }\right)$ for the address of each selected DM minus one check bit for the relative position). Thus, Hi-CPM2 encodes one additional information bit on a sequence of the same length—$2N_{\ell }$—and with equal (two higher-level DMs) or less (when the two selected addresses are the same) energy.

The performance of the HiDM structure using Hi-CPM or Hi-CPM2 mostly depends on the performance of the DMs in the first layer. For example, let us assume that the goal is to minimize the output energy, that is, to approach the MB distribution. A simple idea is to use small LUTs, with $D_{1}^{\left(1\right)}$ comprising the first $2^{k_{1}}$ lowest-energy sequences of length $N_{1}$, $D_{2}^{\left(1\right)}$ the next $2^{k_{1}}$ lowest-energy sequences, and so on with increasing *ℓ*. However, LUTs can be practically implemented only for a short block length $N_{1}$, hence with a high rate loss. More efficient structures can be obtained by replacing LUTs with ESS. In this case, $D_{\ell }^{\left(1\right)}$ is implemented as an ESS whose sequences have energy $E_{\ell -1}<E\leq E_{\ell }$, with $E_{0}=0$ and $E_{\ell }$ selected as the minimum value that allows to encode $2^{k_{1}}$ sequences. Unfortunately, this method implies a small loss with respect to the LUT-based counterpart, since not all the low-energy sequences contained in each energy interval are addressed. Similar approaches can be used with other types of DMs.

### 4.2. Complexity

The encoding performed in each upper layer of the Hi-CPM structure (for ℓ=2,…,L) is extremely simple: the $k_{\ell }$ input bits defines the binary address (in a block of $N_{\ell }=2_{\ell }^{k}$ DMs) in which the DM $D_{\ell }^{\left(1\right)}$ is used in the place of $D_{\ell -1}^{\left(1\right)}$. The decoding consists in the inverse operation: the binary address of the modified DM (in the block of $N_{\ell }$ DMs) corresponds to the $k_{\ell }$ decoded bits. In practice, no additional complexity or resources are required by the upper layers, and the complexity of the overall Hi-CPM structure reduces to that of the single DMs used in the first layer: LUTs, ESS, CCDM, or others. Similar considerations hold for the Hi-CPM2 structure.

### 4.3. Simulation Setup and Results

The Hi-CPM and Hi-CPM2 have been investigated through simulations. Figure 5a,b, which consider small LUTs in the first layer, show the strenght of the method in a general situation.

Figure 5a reports the rate loss for Hi-CPM (solid lines) and Hi-CPM2 (dashed lines) with different number of layers *L*. The rate loss is reported as a function of the rate. The *L* DMs in the first layer are LUTs designed to contain minimum-energy sequences (as explained in the previous section) with block length $N_{1}=10$, and with amplitudes from an alphabet of order $M_{1}=4$. A variable number of input bits to each layer is considered to change the overall rate. The maximum value considered for $k_{\ell }$ for ℓ≥2 is 4. The line with label L=1 is just the rate loss of the LUT $D_{1}^{\left(1\right)}$, representing the best that can be done with block length $N_{1}=10$. For a wide range of rates, the figure shows that the rate loss of Hi-CPM decreases when increasing the number of layers, though no relevant improvements are observed for L>3. As expected, Hi-CPM2 has a similar behavior but a lower rate loss with respect to Hi-CPM. At most, the rate loss improvement with respect to the use of the single LUT slightly exceeds 0.02, with an improvement of about 17% at rate 0.65, and 20% at rate 1.3.

Figure 5b shows that a similar behavior is obtained also in terms of energy loss with respect to the MB distribution. For example, at rate 1.3, the energy loss of the single-layer DM is 0.73dB, whereas that of the 5-layer Hi-CPM2 is 0.59dB, with a gain of 0.14dB.

Different examples could be made by considering different DM types and lengths on the first layer—for instance, one could start by considering longer DMs with a better performance. The general idea is that, regardless of the initial choice, one can reduce the rate loss and energy loss by using the Hi-CPM or Hi-CPM2 structure, with an almost irrelevant increase in complexity.

## 5. The HI-DM with CCDM: Hierarchical CCDM

In this Section, we describe a Hi-DM structure based on CCDMs—referred to as Hi-CCDM in the following—providing a simple design procedure and investigating the performance of two-layer structures. In general, given a target rate, a CCDM generates sequences of a certain length *N* with a given type n, meaning that each possible sequence has a *constant composition*, with $n_{h}$ symbols of type *h* for h=1,…,M (i.e., the amplitude 2h−1 or a DM $D_{h}^{\left(\ell \right)}$). This CCDM encodes *k* bits, with
(14)$$k=\left\lfloor \log_{2}\left(N!/\left(n_{1}!\ldots n_{M}!\right)\right)\right\rfloor .$$

**Remark** **2.**
*The Hi-CCDM is fixed-to-fixed-length even if the DMs of the same layer have a different number of input bits. The proof for two layers is as follows. Let $k_{1,h}$ indicate the number of input bits to the DM $D_{h}^{\left(1\right)}$. The overall number of input bits is given by the number of bits encoded by the second layer, $k_{2}$, plus the contribution of the first layer. The latter depends on the number of occurrences of each DM $D_{h}^{\left(1\right)}$, times the corresponding number of input bits $k_{1,h}$. Since the second layer is a CCDM with type $n^{\left(2\right)}$, the number of occurrences of each DM from the first layer is always the same, regardless of the specific sequence generated by the upper layer. In particular, for a two-layer structure, the total number of input bits is*
(15)$$k=k_{2}+\sum_{h=1}^{M_{2}}k_{1,h}n_{h}^{\left(2\right)}.$$

*The proof can be easily extended to multiple-layer structures.*


### 5.1. Principle of Operation and Design

The basic structure is the same described in Section 3. However, as explained in Remark 2, in this case it is not necessary that all the DMs of a layer have the same number of input bits. Thus, we will remove this condition and consider, in each layer, several CCDMs with different number of input bits but same number of output bits—the *h*th DM of the *ℓ*th layer, $D_{h}^{\left(\ell \right)}$, will have $k_{\ell ,h}$ input bits and $N_{\ell }$ output bits. For the sake of simplicity, we will consider only two-layer structures.

In the first layer, there are $M_{2}$ CCDMs with the same block length $N_{1}$. The *h*th CCDM of the layer, $D_{h}^{\left(1\right)}$, has $k_{1,h}$ input bits and type $n_{h}^{\left(1\right)}$. The types must all be different to ensure that the CCDMs are disjoint. In the second layer, there is another CCDM with block length $N_{2}$, $k_{2}$ input bits, and type $n^{\left(2\right)}$.

The encoding and decoding are performed according to Algorithms 1 and 2, respectively, with minor modifications required to account for the variable number of input bits to the DMs of a single layer. As far as it concerns the decoding, the DM that generated a given sequence can be found just checking the type of the sequence, as already mentioned in Section 3.4.

The rate and the rate loss of the structure are uniquely defined by the types of the CCDMs involved. For a given rate and parameters N and M, the optimal structure uses the types that minimize the rate (or energy) loss. To simplify this optimization, we arbitrarily fix the CCDM types in the first layer, and optimize only the CCDM type in the second layer. In principle, one could check all the possible CCDM types, evaluate the corresponding rate and rate loss, and pick the best values. However, this simple exhaustive search becomes impracticable as the dimensions involved—$N_{2}$ and $M_{2}$—increase. Thus, we propose a simple heuristic procedure to find (almost) optimal combinations. We start with an initial guess $\bar{n}$ for $n^{\left(2\right)}$. The initial guess should be a proper type, that is, ${\bar{n}}_{h}\in N_{0}$ for $h=1,\ldots ,M_{2}$ and $\sum_{h=1}^{M_{2}}{\bar{n}}_{h}=N_{2}$. For example, an approximation of the uniform distribution may be a good starting point. Then, we consider all the possible types that can be obtained from the initial guess by simultaneously increasing one component by one and decreasing another component by one. These types are at most $M_{2}\left(M_{2}-1\right)$. For each of them, we evaluate the corresponding rate and rate loss. The first follows from Equations (Equation 14) and (Equation 15), while the second is evaluated by computing the output probability distribution and the corresponding entropy, as explained in Section 3.5. Then, we repeat the procedure for all the new types. When a rate is obtained twice, we consider only the type with the smallest rate loss and discard the other one.

The proposed heuristic procedure generates many possible types, each with a different rate and a corresponding rate loss. The procedure is not guaranteed to produce optimal results (the obtained types depend on the number of iterations, starting value, etc.). Nonetheless, as shown in Section 5.3, it is simple and effective, and the values of rate and rate loss obtained in this way are achievable with the corresponding types. The same procedure can be followed also by using the energy loss rather than the rate loss as a performance metric.

### 5.2. Complexity

The complexity of the Hi-CCDM depends on the complexity of the component CCDMs. An efficient CCDM implementation based on arithmetic coding is proposed in Reference [6]. This implementation does not require any storage memory (in contrast to ESS or LUTs), and its complexity depends the block length *N*, number of input bits *k*, and alphabet order *M*. Let c(N,k,M) denote the total cost (e.g., number of operations) required by a generic CCDM with parameters *N*, *k*, and *M*. On the one hand, the cost per output amplitude for a single-layer CCDM is
(16)$$c\left(N_{1},k_{1},M_{1}\right)/N_{1}.$$

On the other hand, the cost per output amplitude for a two-layer Hi-CCDM is
(17)$$\sum_{h=1}^{M_{2}}n_{h}^{\left(2\right)}c\left(N_{1},k_{1,h},M_{1}\right)/\left(N_{1}N_{2}\right)+c\left(N_{2},k_{2},M_{2}\right)/\left(N_{1}N_{2}\right).$$

Assuming that all the DMs have similar complexities (i.e., similar parameters), the first term is approximately equivalent to Equation (Equation 16), while the second term is $N_{2}$ times smaller. In this case, the overall cost per amplitude of the Hi-CCDM structure is comparable with that of each component CCDM, and the additional complexity with respect to the single-layer case is negligible. This is a general rule: as long as the upper-layer CCDMs are not much more complex than the first-layer CCDMs, the overall complexity is approximately equal to the (weighted) average complexity of the single component CCDMs of the first layer.

### 5.3. Simulation Setup and Results

We consider a target rate of 1.59 bits/amplitude. In the single-layer case, considering a block length $N_{1}=32$ and $M_{1}=4$ amplitude levels, the design procedure described in Reference [17] returns the CCDM $D_{1}^{\left(1\right)}$, with type $n_{1}^{\left(1\right)}=\left(13,10,6,3\right)$, rate loss of about 0.23 bits/amplitude, and energy loss of about 1.6 dB, as shown in Figure 6a,b. On the other hand, for the two-layer Hi-CCDM structure, we consider $M_{2}=8$ CCDMs with the same block length $N_{1}=32$ and amplitude levels $M_{1}=4$ on the first layer, and a shorter CCDM with block length $N_{2}=10$ in the second layer. In the first-layer, besides the already mentioned $D_{1}^{\left(1\right)}$ designed for the single-layer structure, we consider the CCDMs with types $n_{2}^{\left(1\right)}=\left(14,10,6,2\right)$, $n_{3}^{\left(1\right)}=\left(16,9,5,2\right)$, $n_{4}^{\left(1\right)}=\left(12,9,7,4\right)$, $n_{5}^{\left(1\right)}=\left(14,11,5,2\right)$, $n_{6}^{\left(1\right)}=\left(13,9,7,3\right)$, $n_{7}^{\left(1\right)}=\left(15,11,6,0\right)$, and $n_{8}^{\left(1\right)}=\left(15,10,5,2\right)$. These CCDMs are arbitrarily selected either by designing them for slightly different target rates or by considering minor variations of $D_{1}^{\left(1\right)}$, with no pretense of optimality. Their rates and rate losses are reported in Figure 6a with red squares.

In this case, the short block length considered for the second-layer CCDM ($N_{2}=10$) allows to perform an exhaustive search to find the optimal types. The rate and rate losses obtained by testing all the possible types $n^{\left(2\right)}$ are indicated by the blue dots in Figure 6a. At the target rate of 1.59, a minimum rate loss of 0.1793 is obtained for $n^{\left(2\right)}=\left(1,3,1,1,2,1,0,1\right)$, with an improvement of about 23% with respect to the single-layer structure. As explained in Section 5.2, this improvement is obtained with negligible additional complexity.

In the same figure, the black line shows the results obtained by avoiding the exhaustive search of all the possible $n^{\left(2\right)}$ types, and using the heuristic procedure described in Section 5.1. In this (simple) case, the procedure obtains all the optimal values but in a significantly shorter time. The design procedure yields not only the Hi-CCDM structure for the desired target rate, but also several other good structures for a range of rates around the target rate.

Finally, we consider a block length of $N_{2}=32$ also for the second-layer CCDM, without changing the CCDMs of the first layer. In this case, the exhaustive procedure is quite burdensome and only the result of the heuristic procedure is shown with a green line. By considering the same block length $N_{1}=N_{2}=32$ on both the first and second layer, the rate loss is further reduced, whereas the overall computational complexity remains comparable to that of a single-layer CCDM with a block length of 32 symbols. At the target rate of 1.59, the minimum rate loss of 0.1622 is obtained for $n^{\left(2\right)}=\left(4,6,7,2,6,2,1,4\right)$.

Figure 6b reports the same results as Figure 6a, but replacing the rate loss with the energy loss with respect to the MB distribution. At the target rate of 1.59, the two-layer Hi-CCDMs with $N_{2}=10$ and $N_{2}=32$ reduce the energy loss compared to the best single-layer CCDM with similar complexity by 0.39dB and 0.52dB, respectively.

Figure 7a,b show similar results, but considering longer CCDMs on both layers ($N_{1}=N_{2}=128$) to achieve lower rate and energy losses. The rate (and energy) loss obtained with single-layer CCDMs designed for different target rates is shown with a dashed black line. On the other hand, the performances of the two-layer Hi-CCDM structures are reported with solid colored lines. Each line corresponds to a different set of CCDMs used in the first layer, with rates and rate losses indicated by the symbols with the same color, while the CCDM type in the second layer is optimized by using the heuristic procedure described above.

In all the cases except one (red color), the CCDM types on the first layer are chosen as a subset of the optimal single-layer CCDMs reported along the black dashed line. Both the blue and the yellow lines use $M_{2}=8$ different CCDMs on the first layer. The yellow line considers CCDMs with closer rates, obtaining a better performance (lower rate or energy loss) in a smaller rate interval. On the other hand, the blue line considers CCDMs spanning a wider rate range, obtaining a slightly worse performance over a wider rate interval. In both cases, the rate and energy losses are reduced with respect to the optimal single-layer CCDMs with similar complexity (the black dashed line).

A better performance can be obtained by increasing the number of CCDMs on the first layer to $M_{2}=24$ (green line). In this case, all the available optimal single-layer CCDMs are used (green symbols). Finally, to further improve the performance around the target rate, the same number of CCDMs, $M_{2}=24$, is considered, but spanning a narrower range of rates (red line). In this case, also some non-optimal CCDMs in the desired range of rates must be considered (red symbols). These non-optimal CCDMs are simply obtained by minor variations of the optimal types. The full list of CCDM types used in this case is reported in Table 1. At the rate $R_{0}=507/320\approx 1.59$, a rate loss reduction of about 34% and an energy loss reduction of about 0.18dB are obtained.

Finally, by comparing the results obtained in this section in terms of rate loss and in terms of energy loss, we can observe similar but not identical behaviors. In fact, not all the component CCDMs considered in the first layer are designed to provide the best approximation to the MB distribution. Therefore, the two performance metrics are not fully equivalent.

## 6. The HI-DM with LUT: Hierarchical LUT

This section considers a Hi-DM structure based on the combination of several LUTs, referred to as Hi-LUT. The LUTs are designed to minimize the average output energy for a given structure, implying that the distribution of the output symbols approaches the MB distribution for long block lengths. The use of LUTs makes this approach extremely convenient in terms of computational complexity, as almost no operations are required but LUT accesses. On the other hand, storing all the LUTs may require significant memory resources. Thus, the total required memory will be considered as the relevant cost parameter in this case.

After describing the LUT construction procedure and computing the overall memory required to store all the LUTs, we draw a parallel with the Hi-DM structure described in Yoshida et al. [9,12], investigate the performance of Hi-LUT structures with several layers in terms of rate and energy loss, and show how the performance changes when adding a layer or when tuning the rate without modifying the structure.

### 6.1. Lut Construction

The Hi-DM structure, notation, encoding, and decoding follow the general rules given in Section 3, with an example specifically provided in Section 3.1.2. Once the set of parameters M, N, and k that define the structure have been chosen, the entries of the component LUTs are filled according to the following procedure, iterated for ℓ=1,…,L

Generate the $M_{\ell +1}2^{k_{\ell }}$ sequences of length $N_{\ell }$ with minimum mean energy per symbol from the set $D^{\left(\ell -1\right)}=\left\{D_{1}^{\left(\ell -1\right)},D_{2}^{\left(\ell -1\right)},\ldots ,D_{M_{\ell }}^{\left(\ell -1\right)}\right\}$, where $D_{h}^{\left(0\right)}=2h-1$. The energy of the symbol $d_{h}\in D^{\left(\ell -1\right)}$ is denoted as $E_{d_{h}}^{\left(\ell -1\right)}$ and $E_{d_{h}}^{\left(0\right)}=d_{h}^{2}$. The mean energy per symbol of the sequence $d_{1},\ldots ,d_{N_{\ell }}$ is obtained by averaging the energy of the corresponding DM in the lower level layer, that is, $\sum_{n=1}^{N_{\ell }}E_{d_{n}}^{\left(\ell -1\right)}/N_{\ell }$.Form the *ℓ*th layer LUT $D^{\left(\ell \right)}=\left\{D_{1}^{\left(\ell \right)},\ldots ,D_{M_{\ell +1}}^{\left(\ell \right)}\right\}$ by combining the $M_{\ell +1}2^{k_{\ell }}$ sequences obtained in the previous step, sorted in increasing order of energy. The *h*th component LUT $D_{h}^{\left(\ell \right)}$ contains the $2^{k_{\ell }}$ sequences in the rows $\left(h-1\right)2^{k_{\ell }}+1,\ldots ,h2^{k_{\ell }}$, with $h=1,\ldots ,M_{\ell +1}$, similarly to the first layer of Figure 2b.Evaluate the average energy per symbol of the DMs obtained in previous step,
(18)$$E_{h}^{\left(\ell \right)}=\sum_{r=\left(h-1\right)2^{k_{\ell }}+1}^{h2^{k_{\ell }}}\sum_{n=1}^{N_{\ell }}E_{d_{r,n}}^{\left(\ell -1\right)}/\left(2^{k_{\ell }}N_{\ell }\right),$$
for $h=1,\ldots ,M_{\ell +1}$, with $d_{r}$ being the sequence in the *r*th row of $D^{\left(\ell \right)}$ and $d_{r,n}$ its components. For example, in the structure in Figure 2b the average energy per symbol in $D^{\left(2\right)}$ is $E^{\left(2\right)}=11.7986$.

The *ℓ*th-layer DM can be saved a single LUT $D^{\left(\ell \right)}$. In this case, the step 9 in the Algorithm 1 becomes: read the symbol sequence in the row $\left(j-1\right)2^{k_{\ell }}+1+B$ of $D^{\left(\ell \right)}$, where *B* is the address corresponding to the bit sequence $b_{t}$, and *j* is the index representing $D_{j}^{\left(\ell \right)}.$

It can happen that, in the *ℓ*th layer, only some of the DMs available in the alphabet are actually selected by the design procedure—for instance, $D_{1}^{\left(\ell -1\right)},\ldots ,D_{h}^{\left(\ell -1\right)}$, with $h<M_{\ell }$. This means that the remaining DMs $D_{h+1}^{\left(\ell -1\right)}$,...,$D_{M_{\ell }}^{\left(\ell -1\right)}$ never occur in the sequences that compose the *ℓ*th layer DM. In this case, we can simply avoid to memorize the corresponding LUTs in the lower layer. As a consequence, the previously defined M contains just an upper bound to the effective number of levels used in each layer. The latter can be easily obtained by checking what DMs are actually selected by the design procedure. Thus, in the rest of the paper, we will always consider the effective M.

### 6.2. Complexity and Memory

As anticipated, the Hi-LUT structure requires only LUT accesses and no other operations. Therefore, the complexity of the structure is measured in terms of total memory required to store all the LUTs.

At the encoding side, each layer comprises $M_{\ell +1}$ LUTs; each LUT has $2^{k_{\ell }}$ entries; each LUT entry contains a sequence of $N_{\ell }$ symbols; each symbol belongs to an $M_{\ell }$-ary alphabet and is represented by $m_{\ell }=\log_{2}M_{\ell }$ bits. Thus, the total memory (in bits) required to save all the LUTs of the Hi-DM structure is
(19)$$\sum_{\ell =1}^{L}M_{\ell +1}2^{k_{\ell }}N_{\ell }m_{\ell }.$$

Depending on the desired implementation, some encoding operations can be performed in parallel. For instance, in the *ℓ*th layer, the set of DMs $D^{\left(\ell \right)}$ is used $T_{\ell }$ times to encode $T_{\ell }$ blocks of $k_{\ell }$ bits. These $T_{\ell }$ encoding operations can be performed in parallel by storing $T_{\ell }$ copies of $D^{\left(\ell \right)}$. In this case, if a full parallelization of each layer is considered, the total required memory (in bits) becomes
(20)$$\sum_{\ell =1}^{L}T_{\ell }M_{\ell +1}2^{k_{\ell }}N_{\ell }m_{\ell }.$$

The latter equation corresponds to the memory required by the Hi-DM structure proposed in References [9,12].

In the following, in order to provide a fair comparison with other DM approaches (e.g., ESS), we will consider the basic memory requirement given by Equation (Equation 19), unless otherwise specified. In fact, a similar multiplication of the required memory would take place also when considering a straightforward parallelization of the other DM approaches.

At the decoding side, the required memory depends on the implementation, as already discussed in Section 3.4. For example, if the inverse LUTs are implemented by index mapping to avoid any lookup operation, the total required memory is $\sum_{\ell =1}^{L}M_{\ell }^{N_{\ell }}\left(k_{\ell }+m_{\ell +1}\right)$, which is slightly larger than Equation (Equation 19). Similarly to the encoding, also the decoding can be parallelized, requiring a total memory of $\sum_{\ell =1}^{L}T_{\ell }M_{\ell }^{N_{\ell }}\left(k_{\ell }+m_{\ell +1}\right)$ bits, in accordance with References [9,12]. However, the total memory required at the decoding side can be reduced by employing a memory-efficient implementation of the LUT, as mentioned in Section 3.4.

### 6.3. Comparison with Yoshida et al.

At first sight, the Hi-DM structure proposed in this work and the one proposed in References [9,12] seem to be quite different and based on different working principles. However, it can be shown that the Hi-LUT structure presented in this section becomes equivalent to the Hi-DM structure in References [9,12] when the LUTs are fully parallelized and are filled to minimize the mean energy per symbols according to the procedure detailed in Section 6.1. The analogy between the two structures and the corresponding notations are clarified below.

The block length $N_{\ell }$ corresponds to the term $t_{\ell -1}$ in References [9,12]. Indeed, the block length $N_{\ell }$ is the number of symbols from the ℓ−1 layer, produced by the *ℓ* layer, which is equivalent to the number of copies of the ℓ−1 layer connected to each copy of the *ℓ* layer in an hardware implementation, denoted as $t_{\ell -1}$ in References [9,12]. The parameter $T_{\ell }$ has the same meaning in both manuscripts.The number of levels on each layer, $M_{\ell }$, is related to the number of shaped bits entering the lower-level LUTs, indicated as $r_{\ell -1}$ in References [9,12]. Indeed, $r_{\ell -1}$ is equivalent to $m_{\ell }=\log_{2}M_{\ell }$, the number of bits required to represent the symbols from the alphabet of order $M_{\ell }$.The number of input bits to each layer, $k_{\ell }$, is denoted as $s_{\ell }$ in References [9,12].In our approach, the DMs used in the *ℓ*th layer are determined by the probabilistically shaped symbols coming from the upper layer. This is equivalent to add the binary representations of these symbols coming from the upper layer as prefixes to the input bits to the lower-layer LUTs, as done in References [9,12].

Following this analogy, the 7-layer structure proposed in References [9,12] becomes M=(4,64,64,64,64,64,64), N=(5,2,2,2,2,2,2), and k=(3,5,5,5,5,5,5), and is reported in second-last line of Table 2.

### 6.4. Simulation Setup and Results

Here we analyze the performance of the Hi-LUT structure described in this section, considering a target rate $R_{0}=507/320\approx 1.58$ and an output alphabet of $M_{1}=4$ amplitudes, as in References [9,12].

Figure 8a,b compare the performance of several Hi-LUT structures with different number of layers (shown with symbols), with that of a single-layer LUT (red line) and that of an ESS-based DM (black line).

In particular, Figure 8a reports the rate loss versus the encoding memory, whereas Figure 8b reports the energy loss versus the encoding memory. In this case, since all the structures are built to minimize the mean energy per symbol, the two performance metrics yield equivalent results. The encoding memory required by the Hi-LUT (and single-layer LUT) structures is given by Equation (Equation 19). On the other hand, the ESS encoding memory is that required to store the trellis [11]. Indeed, as mentioned in the Introduction, the ESS approach uses a trellis structure to map uniformly distributed sequences of *k* bits—that is, indexes from 1 to $2^{k}$—to sequences of *N* amplitudes with energy $E\leq E_{\max}$, where the energy bound $E_{\max}$ is selected as the minimum value for which at least $2^{k}$ sequences are available. Often, a larger number of sequences falls within $E_{\max}$. In this case, the optimal shaping is obtained by discarding some of the sequences with $E=E_{\max}$. This can be done by storing two trellises, one for the sequences with $E<E_{\max}$, and one for those with $E=E_{\max}$. In the following figures, the label ESS indicates this optimal method, and the required memory accounts for both the trellises.

For the rate $R_{0}$ and a given number of layers *L*, we consider all the possible Hi-LUT structures that require a *reasonable* amount of memory and construction effort, and we plot only the best results. For L=7, only two Hi-LUT configurations are shown, the one proposed in References [9,12] and a slightly modified one, respectively reported in the second-last line and last line of Table 2. For the former, two different values of required memory are reported in the figure: one given by Equation (Equation 20) and corresponding to the fully parallelized implementation considered in References [9,12]; the other given by Equation (Equation 19) and labeled as “generalized”. The latter structure achieves a rate loss of 0.0404, with a 33% improvement with respect to the equivalent-memory ESS (which has N=28 and rate loss of 0.0603) and a slight improvement with respect to References [9,12].

More in general, Figure 8a shows that, for a given target rate, many different Hi-LUT structures are available with a different trade-off between performance and required memory. In particular, for a rate loss higher than 0.035, we were able to find several Hi-LUT structures with better performance than the equivalent-memory ESS. Some of the most interesting structures are reported in Table 2, along with their effective rate $R_{\mathrm{eff}}$, rate loss $R_{\mathrm{loss}}$, energy loss $E_{\mathrm{loss},\mathrm{dB}}$, and required memory. We conjecture that additional improvements and lower rate and energy losses can be achieved with additional layers, provided that an efficient (non-exhaustive) optimization strategy is devised. This is, however, beyond the scope of this paper.

Note that, in contrast to LUTs, the ESS approach requires also some non-trivial encoding and decoding operations. This is an additional cost for ESS that should be taken into account when comparing it to Hi-LUT structures. Therefore, though the best trade-off between required memory and computational cost may depend on several factors, we can clearly state that, for the same performance and required memory, Hi-LUTs are preferable to ESS.

Finally, Figure 9a,b reports the same results shown in the previous figures—rate loss and energy loss, respectively, for L=1,…,5—but as a function of the overall block length *N*. Different colors of the symbols correspond to different ranges of encoding memory Equation (Equation 19). The figures also report the performance for sphere shaping (e.g., obtained by ESS or LUT, as the implementation algorithm is not relevant in this case) and CCDM. Though the overall block length does not provide a direct indication of the implementation complexity, this figures reveal how efficiently the hierarchical structures select sequences of a given length with respect to sphere shaping, which is the most efficient packing solution. The figures show that (i) CCDM requires a much longer block length compared to sphere shaping to obtain the same performance [7], (ii) the Hi-LUT performance approaches that of sphere shaping for short block lengths, but deviates from it for longer block lengths, (iii) many Hi-LUT structures perform better than CCDM for the same block length, and (iv) the Hi-LUT rate loss decreases and approaches sphere shaping—the optimal packing solution—when a larger memory is allowed.

### 6.5. Rate and Rate Loss Propagation Adding an External Layer

This section analyzes how the rate and the rate loss change when adding an external layer to an existing Hi-LUT structure. In particular, the structure reported in Table 2 with L=4 layers, M=(4,64,64,64,64), N=(6,2,3,3), and k=(5,5,8,12) is considered.

As follows from Equation (Equation 9), adding a layer to a given structure increases its rate. Indeed, if $R^{\left(L\right)}$ is the rate of the structure with *L* layers, then
(21)$$R^{\left(L+1\right)}=R^{\left(L\right)}+\frac{R_{L+1}}{\prod_{h=1}^{L}N_{h}}.$$

Figure 10a shows the rate and the corresponding rate loss that is obtained by adding an external layer to the four-layer structure. The figure shows the behavior for different values of $N_{5}$ and different rates, obtained with different $k_{5}$. As expected, the rate loss is lower for larger $N_{5}$. Interestingly, for fixed $N_{5}$, the rate loss decreases down to an optimal value when increasing the rate (i.e., increasing $k_{5}$), reaches an optimal value, and then increases again. This is due to the trade-off between rate and entropy, both increasing when $k_{5}$ increases. Indeed, when $k_{5}$ starts to increase, the rate loss diminishes because additional bits are encoded. However, since the $D_{1}^{\left(5\right)}$ outputs $2^{k_{5}}$ sequences, when $k_{5}$ is too large, the DMs from the lower levels with large entropy (energy) are selected many more times, badly affecting the overall rate loss. This happens earlier for smaller $N_{5}$ because there are less possible combinations—they are $M_{5}^{N_{5}}$—-and, thus, high-entropy DMs are chosen more often.

### 6.6. Tuning the Rate of a Fixed HI-LUT Structure: Rate Loss and Granularity

In this section we consider a fixed Hi-LUT structure, and show how the rate and the rate loss change when modifying the vector $k=(k_{1},\ldots ,k_{L})$, that is, the number of bits encoded on each layer. The overall rate of the Hi-LUT structure is given by Equation (Equation 9). After multiplying and dividing by the total number of output amplitudes $N=\prod_{\ell =1}^{L}N_{\ell }$, Equation (Equation 9) can be rewritten as
(22)$$R=\frac{k}{N}=\frac{\sum_{\ell =1}^{L}k_{\ell }T_{\ell }}{N},$$
where $T_{\ell }=\prod_{h=\ell +1}^{L}N_{h}$ is the number of times the *ℓ*th layer is used. According to Equation (Equation 22), the rate of the structure can be increased or decreased by respectively increasing or decreasing the number of input bits to one of the layers. In general, changing the vector k changes the structure, since each layer $D^{\left(\ell \right)}$ uses the first $M_{\ell }2^{k_{\ell }}$ sequences with minimum energy. However, given a Hi-LUT structure with a certain k, it is possible to *decrease* the number of input bits—hence, to decrease the overall rate—without changing the structure and the content of the LUTs. In particular, one can use any $\bar{k}$, with $\bar{k_{\ell }}\leq k_{\ell }$ for ℓ=1,…,L, by simply changing the address of the sequences. For instance, assuming that all the LUTs of the *ℓ*th layer are saved as a single LUT $D^{\left(\ell \right)}$, let *B* be the address corresponding to a certain input bit sequence and *j* the index representing the particular LUT $D_{j}^{\left(\ell \right)}$ to be used for the encoding. When the original vector k is considered, the corresponding output sequence is found in the row $\left(j-1\right)2^{k_{\ell }}+1+B$ of $D^{\left(\ell \right)}$, as explained at the end of Section 6.1. On the other hand, when the modified vector $\bar{k}$ is considered, the output sequence is found in the row $\left(j-1\right)2^{{\bar{k}}_{\ell }}+1+B$. In this case, some of the sequences stored in the LUTs will simply not be used. Therefore, one can implement a Hi-LUT structure designed for a fairly high k (and rate), and then use the same structure also for lower rates by considering a lower number of input bits on one or more layers.

As an example, we consider the structure with L=5 layers reported in Table 2 with M=(4,64,…,64) and N=(6,2,3,3,3). As a starting point, we consider k=(5,5,8,8,12), yielding the rate 1.58. Figure 10b shows the rate loss versus rate, when changing one $k_{\ell }$ at a time or when changing all of them by the same amount −2,−1,0,or+1 (last line). As the rate changes according to Equation (Equation 22), acting on the lower layers (where $T_{\ell }$ is larger) gives a wider tuning range, whereas acting on the higher layers (where $T_{\ell }$ is smaller) gives a finer granularity. However, acting on the higher layers causes also a faster increase of the rate loss (for the same desired rate). Therefore, to obtain a certain desired rate, it is better to change the number of input bits on the lower layers, moving to the higher ones only if a finer tuning is desired. For instance, in the example of Figure 10b, a similar rate R≈1.4 can be obtained by reducing $k_{1}$ by one bit, $k_{2}$ by two bits, or $k_{3}$ by six bits. The lowest rate loss is obtained in the first case, and the highest one in the last case. Therefore, if the desired rate is R=1.4, acting on $k_{1}$ is the obvious choice, while some higher layer (e.g., $k_{4}$ or $k_{5}$) can be considered if a more accurate tuning is required.

## 7. System Performance

This Section analyzes the performance of a system employing the proposed Hi-DM structures for the implementation of probabilistic amplitude shaping (PAS). The performance is evaluated for the AWGN channel and is given in terms of generalized mutual information (GMI), a common measure to predict the amount of reliable information that can be transmitted through the channel, given that an ideal FEC and bit-wise decoding are used [10,18].

Figure 11a reports the GMI as a function of the SNR for a uniform-64QAM (U-64QAM) constellation and for a PAS-64QAM constellation with different DM implementations. In particular, we compare the Hi-CCDM with two layers, $M_{2}=28$ and $N_{1}=N_{2}=128$, defined in Table 1 (solid blue) with the single-layer CCDM with comparable complexity, that is, with block length N=128 (dashed blue). Moreover, we compare the seven-layer Hi-LUT structure defined in the last line of Table 2 (solid red) with the ESS with N=28 (dashed red), which requires the same storage memory (but has higher computational complexity). All the DMs structures are optimized for a (one-dimensional) target rate $R_{0}=507/320$ bits per amplitude, which corresponds to a gross rate of $R_{g}=5.1688$ bits per QAM symbol. For example, assuming an ideal FEC with rate $r_{\mathrm{FEC}}=3/4$, the information rate will be $R_{g}-\left(1-r_{\mathrm{FEC}}\right)\log_{2}\left(64\right)=3.6687$ bits per QAM symbol [2]. Finally, the ultimate performance achievable with an ideal MB shaping with i.i.d. symbols (optimized for the same target rate) is reported as a reference. Figure 11b reports the same results as Figure 11a, but focusing on the performance around the target rate for which all the PAS systems are designed.

In general, the results are consistent with those obtained in the previous sections in terms of rate loss and energy loss. All the considered PAS-64QAM systems provide some shaping gain with respect to the U-64QAM system, achieving the same GMI at a lower SNR. The ultimate shaping gain provided by i.i.d. MB symbols is 0.89dB. The two-layer Hi-CCDM structure provides a shaping gain of 0.56dB, 0.19dB more than the corresponding single-layer CCDM structure with same complexity. The seven-layer Hi-LUT structure provides a shaping gain of 0.61dB, 0.12dB more than the corresponding single-layer ESS structure with same required storage memory. Note, however, that the ESS requires performing also many encoding and decoding operations, which are avoided in the Hi-LUT configuration.

This example demonstrates that Hi-DM structures may provide a significant advantage over their single-layer counterparts, achieving higher shaping gains with similar computational complexity and/or storage memory (or, conversely, same gains with lower complexity and memory). Moreover, the specific Hi-DM configurations considered in this work are just an example of some good structures that can be obtained with the hierarchical approach, with no claim of optimality. In fact, better configurations might be possibly devised, for instance by considering Hi-CCDM structures with more than two layers, or with a more systematic optimization of the seven-layer Hi-LUT structure.

## 8. Conclusions

In this work, we have presented a general Hi-DM concept for the implementation of PAS, in which several short DMs are combined in a hierarchical structure to form a longer DM. The set of DMs collected in each layer forms a “virtual modulation alphabet” for the DMs of the upper layer, which encode information on the particular sequence of DMs used in the lower layer. The goal of the Hi-DM structure is that of obtaining the good performance of a long DM with the low complexity of many small DMs.

First, we have described the structure, the properties, and the encoding and decoding operations of a generic Hi-DM configuration. Then, we have proposed three particular types of Hi-DM, obtained by combining different types of DM—Hi-CPM, in which the information is encoded on the *position* of a higher-energy DM with respect to a lower-energy DM; Hi-CCDM, based on the hierarchical combination of *constant-composition* DMs; and Hi-LUT, based on the hierarchical combination of *lookup tables*. For each Hi-DM type, we have provided some practical examples and investigated the performance in terms of rate loss and energy loss compared to the corresponding single-layer structures. Moreover, we have proposed a practical design procedure for the two-layer Hi-CCDM structure, and we have investigated how the rate of a Hi-LUT can be tuned without changing the structure, and how its performance changes by adding one more layer. Finally, we have compared the performance of the proposed Hi-DM structures and of the equivalent-complexity single-layer structures in terms of GMI achievable over the AWGN channel. In particular, a two-layer Hi-CCDM, with block lengths $N_{1}=N_{2}=128$, provides an improvement of 0.19 dB with respect to the single-layer CCDM with same computational complexity; a seven-layer Hi-LUT provides an improvement of 0.12 dB with respect to the single-layer ESS with same memory requirement and much lower computational complexity—a single-layer LUT with comparable performance being not even implementable in this case due to the huge memory requirement.

We believe that PAS will be an essential technique to improve the performance and the flexibility of next generation optical systems, and that the proposed Hi-DM concept is an effective approach for its implementation. With this approach, the gap to the Shannon limit can be significantly reduced with limited hardware requirements. Note, however, that the achievable information rate depends also on the alphabet size and on the FEC code length, meaning that progresses in transceiver technology and error correction will also play an important role in closing the gap.

Further investigations and possible improvements include (i) the comparison of the end-to-end performance obtained by Hi-DM and single-layer DM including the FEC; (ii) the investigation of the error propagation at the inverse DM stage; (iii) the design of more efficient Hi-DM structures with many layers. For what concerns the last point, increasing the number of layers can generally improve the Hi-DM performance for the same computational complexity (or, equivalently, reduce the complexity for the same performance), as shown in many examples in this paper. However, the design of optimal Hi-DM structures becomes more complicated as the number of layers increases, due to the large number of degrees of freedom. In this work, we have considered a simplified design procedure for Hi-CCDM structures with only two layers, and an exhaustive-search design procedure for Hi-LUT structures, which becomes quite cumbersome for more than five layers. In fact, for seven layers we did not perform an exhaustive search but only presented two “good” structures. Therefore, another important research direction is to develop a fast and effective design procedure to find optimal (e.g., in terms of rate or energy loss) Hi-DM structures with many layers and specific constraints in terms of complexity, memory, and block length.

## Figures and Tables

**Figure 1 entropy-22-00958-f001:**
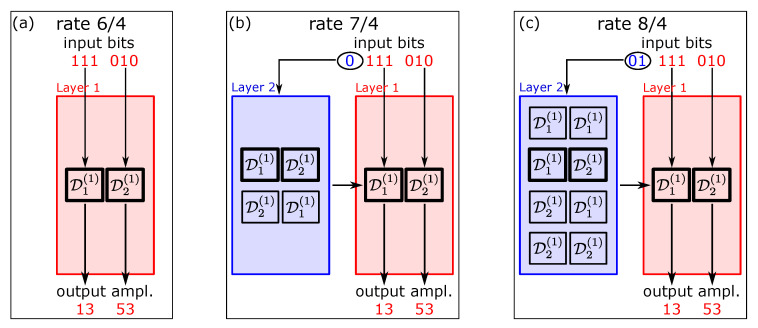
Toy example 1: two distribution matchers (DMs) are combined (**a**) independently, in a deterministic sequence; (**b**) in a hierarchical DM (Hi-DM) structure with 1 additional input bit; (**c**) in a Hi-DM structure with 2 additional input bits.

**Figure 2 entropy-22-00958-f002:**
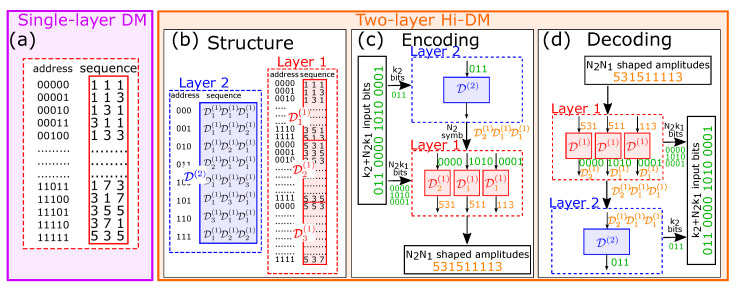
Toy example 2: (**a**) minimum-energy single-layer look up table (LUT) with k=5, N=3, and rate R=5/3; (**b**) Hi-DM structure with two layers, k=(4,3), N=(3,3), M=(4,3), and same rate R=15/9≈1.67 as (**a**); example of (**c**) encoding and (**d**) decoding for the input bit sequence 011000010100001 and the corresponding output sequence 531511113.

**Figure 3 entropy-22-00958-f003:**
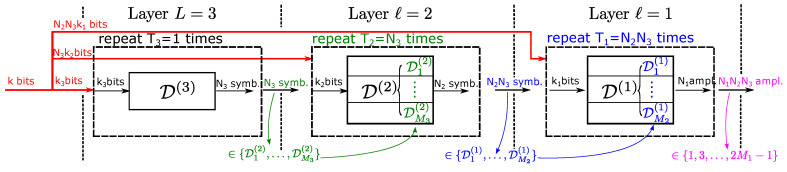
Generic Hi-DM structure with L=3 layers.

**Figure 4 entropy-22-00958-f004:**
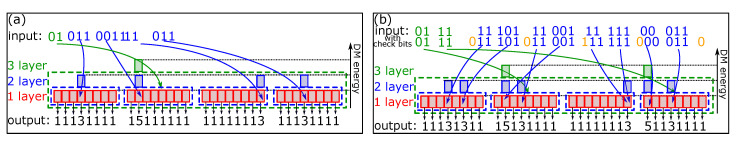
(**a**) Hierarchical code position modulation (Hi-CPM) and (**b**) Hi-CPM2 with 3 layers, amplitudes $\left\{1,3,5,7\right\}$ on the first layer, and k=(0,3,2).

**Figure 5 entropy-22-00958-f005:**
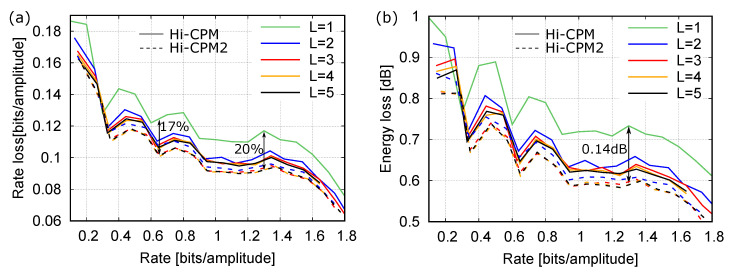
(**a**) Rate loss and (**b**) energy loss versus rate for Hi-CPM and Hi-CPM2 with different number of layers *L*. The *L* DMs in the first layer are LUTs with $N_{1}=10$ and $M_{1}=4$.

**Figure 6 entropy-22-00958-f006:**
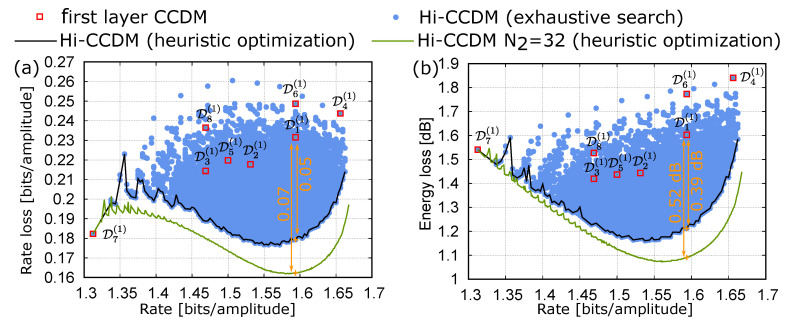
(**a**) Rate loss and (**b**) energy loss versus rate for Hi-CCDM with $M_{1}=4$, $M_{2}=8$, $N_{1}=32$, and $N_{2}=10$ (except the green line, obtained with $N_{2}=32$).

**Figure 7 entropy-22-00958-f007:**
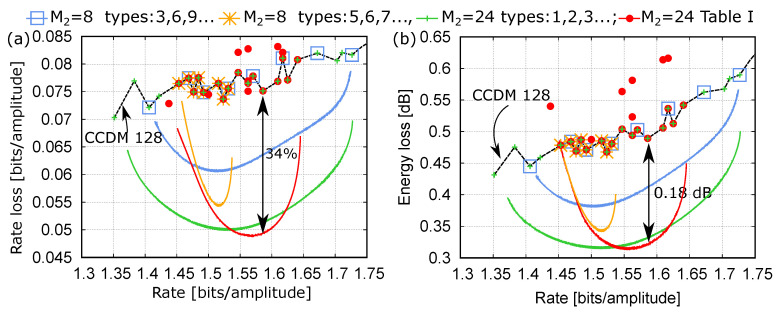
(**a**) Rate loss and (**b**) energy loss versus rate for Hi-CCDM with $N_{1}=N_{2}=128$, and $M_{1}=4$. Each solid line denotes a Hi-CCDM obtained with a different set of constant composition DMs (CCDMs) on the first layer (whose rate/energy loss and rate are indicated by the symbols with the same color). For comparison, the performance of the single-layer CCDM with N=128 is shown with a dashed line.

**Figure 8 entropy-22-00958-f008:**
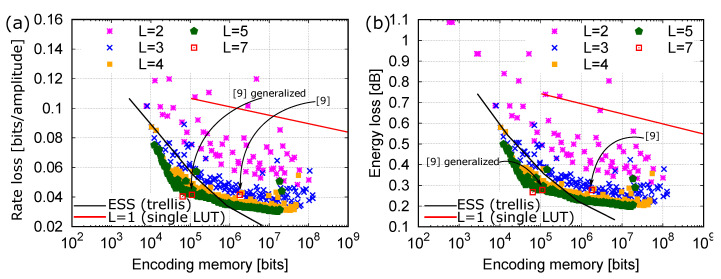
(**a**) Rate loss and (**b**) energy loss versus encoding memory Equation (Equation 19) for Hi-LUTs with a variable number of levels *L* and maximum $M_{\ell }=64$ for ℓ=2,…,L. For L=7, only the Hi-LUT equivalent to the structure proposed in References [9,12] (with and without parallelization) and a different Hi-LUT with slightly better performance are considered.

**Figure 9 entropy-22-00958-f009:**
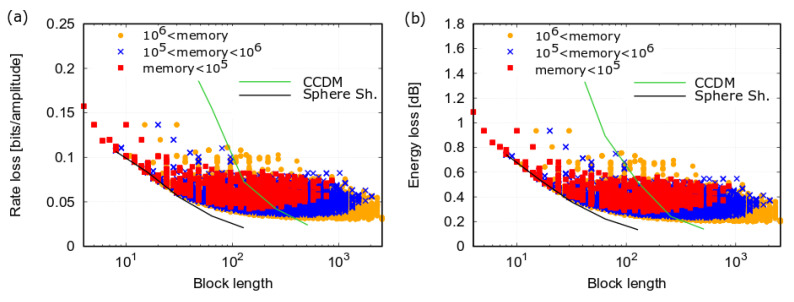
(**a**) Rate loss and (**b**) energy loss versus block length for Hi-LUT with L=1,…,5 layers, $M_{max}=64$, and different maximum encoding memory (without parallelization).

**Figure 10 entropy-22-00958-f010:**
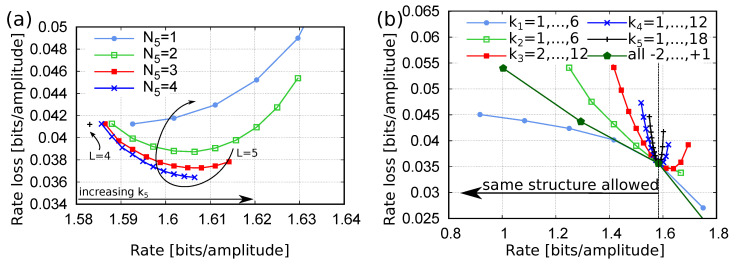
(**a**) Rate loss versus rate obtained by adding one external layer to a four-layer structure. (**b**) Rate loss and rate granularity for a fixed Hi-LUT structure with five layers.

**Figure 11 entropy-22-00958-f011:**
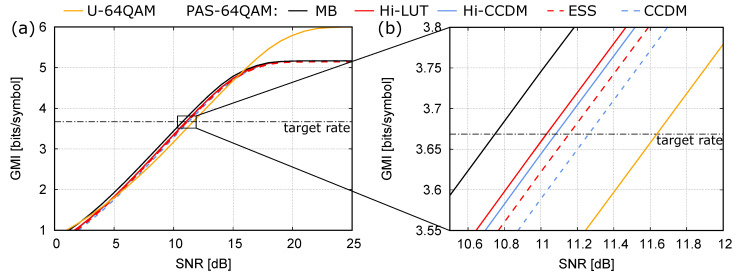
Generalized mutual information (GMI) versus signal to noise ratio (SNR) for a uniform 64-QAM system and for PAS-64QAM systems employing different types of DMs (single-layer or hierarchical). The overall behavior is shown in (**a**), while (**b**) focuses on the performance at the target rate of 3.6687 bits per QAM symbol.

**Table 1 entropy-22-00958-t001:** CCDM types corresponding to the red symbols in Figure 7a,b, used in the first layer of the corresponding Hi-CCDM structure. The last line reports the CCDM type in the second layer that yields a rate $R_{0}\approx 1.59$.

$n_{1}^{\left(1\right)}$ = (68,41,15,4)	$n_{9}^{\left(1\right)}$ = (62,42,19,5)	$n_{17}^{\left(1\right)}$ = (68,38,17,5)
$n_{2}^{\left(1\right)}$ = (67,41,16,4)	$n_{10}^{\left(1\right)}$ = (62,42,18,6)	$n_{18}^{\left(1\right)}$ = (66,38,17,7)
$n_{3}^{\left(1\right)}$ = (66,42,16,4)	$n_{11}^{\left(1\right)}$ = (62,41,19,6)	$n_{19}^{\left(1\right)}$ = (66,37,18,7)
$n_{4}^{\left(1\right)}$ = (66,41,17,4)	$n_{12}^{\left(1\right)}$ = (61,41,20,6)	$n_{20}^{\left(1\right)}$ = (64,35,20,9)
$n_{5}^{\left(1\right)}$ = (65,42,17,4)	$n_{13}^{\left(1\right)}$ = (60,41,20,7)	$n_{21}^{\left(1\right)}$ = (64,36,19,9)
$n_{6}^{\left(1\right)}$ = (65,41,17,5)	$n_{14}^{\left(1\right)}$ = (59,41,21,7)	$n_{22}^{\left(1\right)}$ = (66,36,19,7)
$n_{7}^{\left(1\right)}$ = (64,42,17,5)	$n_{15}^{\left(1\right)}$ = (58,42,21,7)	$n_{23}^{\left(1\right)}$ = (65,37,20,6)
$n_{8}^{\left(1\right)}$ = (64,41,18,5)	$n_{16}^{\left(1\right)}$ = (58,41,21,8)	$n_{24}^{\left(1\right)}$ = (69,37,20,2)
$n^{\left(2\right)}=\left(2,3,4,4,5,7,8,8,6,9,8,10,8,5,6,4,6,3,7,2,2,3,7,1\right)$		

**Table 2 entropy-22-00958-t002:** Example of structures with good trade-off between rate loss and required memory. The second-last line is the structure proposed in References [9,12].

*L*	*M*	*N*	*k*	$R_{\mathrm{eff}}$	$R_{\mathrm{loss}}$	$E_{\mathrm{loss},\mathrm{dB}}$	Equation (Equation 19)	Equation (Equation 20)
3	4,8,16	5,4,4	6,5,11	1.5875	0.0627	0.4278	$4.4\;\times \;10^{4}$	$1.4\times 10^{5}$
3	4,128,128	12,3,3	14,11,12	1.5833	0.0359	0.2331	$5.6\;\times \;10^{7}$	$4.7\times 10^{8}$
3	4,64,64	10,4,4	12,12,14	1.5875	0.0351	0.2303	$1.2\;\times \;10^{7}$	$1.1\times 10^{8}$
3	4,64,64	12,4,4	15,12,16	1.5833	0.0338	0.2234	$5.8\;\times \;10^{7}$	$8.3\times 10^{8}$
4	4,64,64,64	5,2,2,2	3,5,5,9	1.5750	0.0584	0.3864	$6.0\;\times \;10^{4}$	$2.0\times 10^{5}$
4	4,64,64,64	6,2,3,3	5,5,8,12	1.5833	0.0412	0.2716	$4.2\;\times \;10^{5}$	$1.6\times 10^{6}$
4	4,64,64,64	9,4,4,4	10,13,12,17	1.5851	0.0312	0.2022	$2.3\;\times \;10^{7}$	$3.1\times 10^{8}$
4	4,64,64,64	12,4,4,4	15,12,12,17	1.5846	0.0300	0.1956	$6.6\;\times \;10^{7}$	$3.4\times 10^{9}$
5	4,64,…,64	6,2,2,2,3	5,5,4,4,12	1.5833	0.0412	0.2716	$1.5\;\times \;10^{5}$	$1.1\times 10^{6}$
5	4,64,…,64	6,2,3,3,3	5,5,8,8,12	1.5833	0.0356	0.2379	$7.1\;\times \;10^{5}$	$5.6\times 10^{6}$
5	4,64,…,64	10,4,…,4	12,12,10,11,12	1.5844	0.0303	0.1969	$1.6\;\times \;10^{7}$	$1.8\times 10^{9}$
7	4,64,…,64,13	5,2,…,2	3,5,…,5	1.5844	0.0416	0.2789	$1.1\;\times \;10^{5}$	$1.8\times 10^{6}$
7	4,55,64,…,64	5,2,…,2	4,4,4,4,3,3,9	1.5844	0.0404	0.2693	$6.4\;\times \;10^{4}$	$1.3\times 10^{6}$

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
