# Peer review of "Hierarchical Distribution Matching for Probabilistic Amplitude Shaping†"

_entropy, 2020, doi:10.3390/e22090958_

Round 1

Reviewer 1 Report

Review of Stella Civelli and Marco Secondini (Entropy MDPI)

As the authors correctly mentioned, the submitted manuscript is an extended version of "Hierarchical distribution matching: a versatile tool for probabilistic shaping.", sent this year to the "Optical Fiber Communications Conference and Exhibition (OFC). IEEE, 2020." It is a better and complete version of the previous one, and indeed a lengthy manuscript. Usually, long manuscripts are hard for the reader to follow; however, the well-organized sections that include a straightforward language make this lengthy manuscript not so hard to read, therefore to follow.

The proposed concept of having a probabilistic amplitude shaping implemented with a cluster of short Hi-DMs seems ingenious, from the theoretical point of view. Still, I have a couple of questions that I should like to see addressed:

a) Do the authors consider that their PAS/Hi-DM concept will not be challenged due to buffer overflow?

b) Do the authors genuinely believe that the Shannon limit could be tangible with PAS/Hi-DM's proposed concept? If yes, could they explain why?

c) Could the authors offer a summarized explanation on how they could get higher gains, with "Hi-CCDM structures with more than two layers or devising an efficient method to find optimal Hi-LUT structures."? In particular, without having enormous practical challenges, for example, with delays?

d) Are the authors taking into consideration the Big O notation when writing the algorithms?

After these questions are addressed, I will recommend the acceptance of the manuscript.

Reviewer 2 Report

The work is strong and the paper is well organized Thank you. My Comments

1) In Abstract - Start a new sentence with (A significant effort has been made to devise efficient DM structures with a good performance and low hardware requirements).

2) The following paragraph at the end of "Introduction" needs to be expanded to show your work in more details. ((In this work, we expand on [13] by providing a detailed description of the Hi-DM structure and by studying some specific configurations, their design, their performance, and their complexity.))

3) Equations (3),(4) and (11) need a reference.

4) The line before Eq. (6) has a superscript (2) at its end..... I prefer to put your explanation directly (not in a footnote).

5) The lines just before subtitle 6.4 Finally, in our notation, the 7-layer structure proposed in [9,12] becomes M = (4, 64, 64, 64, 64, 64, 64), N = (5, 2, 2, 2, 2, 2, 2), and k = (3, 5, 5, 5, 5, 5, 5), and is reported in second-last line of Table 2. Explain how did you get these new values for M, N, and k.

6) In Table 2, use the multiplication symbol (x) NOT (.) because you have numbers. The table caption must be up (not down) the table.

7) I noticed that you compared your results with your results... Please, compare with previously published work (even with one published paper).

8) Ref. 8 needs last page number. Ref. 12 is Archive???? Ref. 13 needs the place where conference was held and exact date (day-month-year).

Round 2

Reviewer 1 Report

Thank you for your detailed answers.

Reviewer 2 Report

I see authors carefully considered all my comments.

Thanks to authors.